# `PathHD`: Efficient Large Language Model Reasoning over Knowledge Graphs via Hyperdimensional Retrieval

## Abstract

Recent advances in large language models (LLMs) have enabled strong reasoning over structured and unstructured knowledge. When grounded on knowledge graphs (KGs), however, prevailing pipelines rely on neural encoders to embed and score symbolic paths, incurring heavy computation, high latency, and opaque decisions, which are limitations that hinder faithful, scalable deployment. We propose a lightweight, economical, and transparent KG reasoning framework, **PathHD**, that replaces neural path scoring with *hyperdimensional computing* (HDC). PathHD encodes relation paths into block-diagonal *GHRR* hypervectors, retrieves candidates via fast cosine similarity with Top-$K$ pruning, and performs a *single* LLM call to produce the final answer with cited supporting paths. Technically, PathHD provides an order-aware, invertible binding operator for path composition, a calibrated similarity for robust retrieval, and a one-shot adjudication step that preserves interpretability while eliminating per-path LLM scoring. Extensive experiments on WebQSP, CWQ, and the GrailQA split show that `PathHD` (i) achieves comparable or better Hits@1 than strong neural baselines while using *one* LLM call per query; (ii) reduces end-to-end latency by **40–60%** and GPU memory by **3-5×** thanks to encoder-free retrieval; and (iii) delivers faithful, path-grounded rationales that improve error diagnosis and controllability. These results demonstrate that HDC is a practical substrate for efficient KG-LLM reasoning, offering a favorable accuracy-efficiency-interpretability trade-off.

## 1 Introduction

Large Language Models (LLMs) have rapidly advanced reasoning over both text and structured knowledge. Typical pipelines follow a *retrieve–then–reason* pattern: they first surface evidence (documents, triples, or relation paths), then synthesize an answer using a generator or a verifier (Lewis et al., 2020; Press et al., 2022; Yao et al., 2023; Wei et al., 2022; Yao et al., 2024). In knowledge-graph question answering (KGQA), this often becomes *path-based reasoning*: systems construct candidate relation paths that connect the topic entities to potential answers and pick the most plausible ones for final prediction (Sun et al., 2018; Jiang et al., 2022; 2023; 2024; Luo et al., 2023). While these approaches obtain strong accuracy on WebQSP, CWQ, and GrailQA, they typically depend on heavy neural encoders (e.g., Transformers or GNNs) or repeated LLM calls to rank paths, which makes them slow and expensive at inference time—especially when many candidates must be examined.

Figure 1 highlights two recurring issues in KG–LLM reasoning. ❶ **Path–query mismatch:** Order-insensitive encodings, weak directionality, and noisy similarity often favor superficially related yet misaligned paths, blurring the question's intended relation. ❷ **Per-candidate LLM scoring:** Many systems score candidates sequentially, so latency and token cost grow roughly linearly with set size; batching is limited by context/API, and repeated calls introduce instability, yet models can still over-weight long irrelevant chains, hallucinate edges, or flip relation direction. Most practical pipelines first detect a topic entity, enumerate $10 \sim 100$ length-1–4 paths, then score each with a neural model or LLM, sending top paths to a final step (Sun et al., 2018; Luo et al., 2023; Jiang et al., 2024). This hard-codes two inefficiencies: (i) *neural scoring dominates latency* (fresh encoding/prompt per candidate), and (ii) *loose path semantics* (commutative/direction-insensitive encoders conflate *founded_by→CEO_of* with its reverse), which compounds on compositional/long-hop questions.

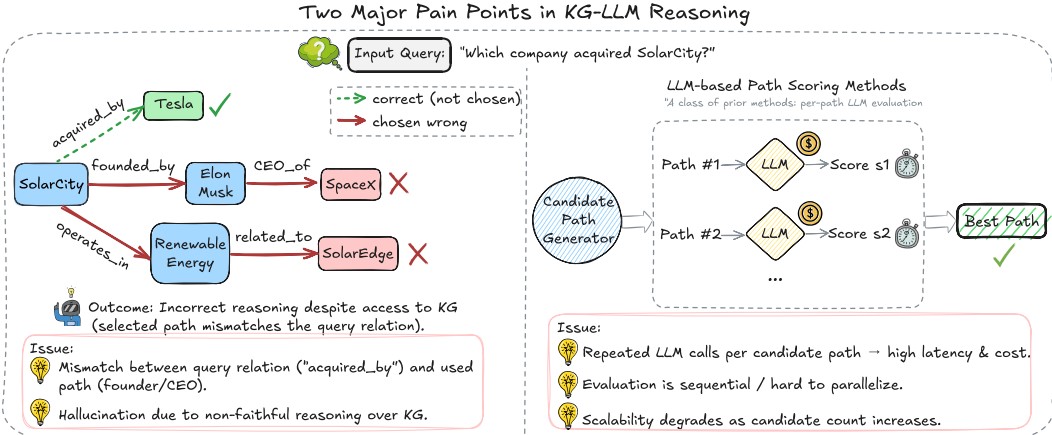

Figure 1: **Two pain points in KG–LLM reasoning.** (Left) The method selects a path that does not match the query relation, leading to wrong answers even with KG access; (Right) per-path LLM scoring incurs high latency and cost because candidates are evaluated one by one.

Hyperdimensional Computing (HDC) offers a different lens: represent symbols as long, nearly-orthogonal *hypervectors* and manipulate structure with algebraic operations such as *binding* and *bundling* (Kanerva, 2009; Plate, 1995). HDC has been used for fast associative memory, robust retrieval, and lightweight reasoning because its core operations are elementwise or blockwise and parallelize extremely well on modern hardware (Frady et al., 2021). Encodings tend to be noise-tolerant and compositional; similarity is computed by simple cosine or dot product; and both storage and computation scale linearly with dimensionality. Crucially for KGQA, HDC supports *order-sensitive* composition when the binding operator is non-commutative, allowing a path like $r_1 \rightarrow r_2 \rightarrow r_3$ to be distinguished from its permutations while remaining a single fixed-length vector. This makes HDC a promising substrate for ranking many candidate paths without invoking a neural model for each one.

Motivated by these advantages, we introduce `PathHD` (Hyper**D**imensional **Path** Retrieval), a lightweight retrieval-and-reason framework for KGQA. First, we map every relation to a block-diagonal unitary representation and encode a candidate path by *non-commutative* Generalized Holographic Reduced Representation (GHRR) binding (Yeung et al., 2024); this preserves order and direction in a single hypervector. In parallel, we encode the query into the same space to obtain a *query hypervector*. Second, we score *all* candidates via cosine similarity to the query hypervector and keep only the top-$K$ paths with a simple, parallel Top-$K$ selection. Finally, instead of per-candidate LLM calls, we make *one* LLM call that sees the question plus these top-$K$ paths (verbalized), and it outputs the answer along with cited supporting paths. In effect, `PathHD` addresses both pain points in Fig. 1: order-aware binding reduces path–query mismatch, and vector-space scoring eliminates per-path LLM evaluation, cutting latency and token cost. Our contributions can be summarized as follows.

❶ **A fast, order-aware retriever for KG paths.** We present `PathHD`, which uses GHRR-based, non-commutative binding to encode relation *sequences* into hypervectors and ranks candidates with plain cosine similarity—no neural encoders and no per-path prompts. This design keeps a symbolic structure while enabling fully parallel scoring with $\mathcal{O}(Nd)$ complexity.

❷ **An efficient one-shot reasoning stage.** `PathHD` replaces many LLM scoring calls with a single, final LLM adjudication over the top-$K$ paths. This decouples retrieval from generation, lowers token usage, and improves wall-clock latency without sacrificing interpretability: the model cites the supporting path(s) it used.

❸ **Extensive validation and operator study.** On WebQSP, CWQ, and GrailQA, `PathHD` achieves competitive Hits@1 with markedly lower cost. An ablation on binding operators shows that our block-diagonal (GHRR) binding outperforms commutative binding and circular convolution, confirming the value of order preservation; additional studies analyze the impact of top-$K$ pruning and latency–accuracy trade-offs.

Rather than proposing a new theory for vector symbolic architectures or hyperdimensional computing, PathHD is aimed at demonstrating that carefully designed HDC representations can serve as a

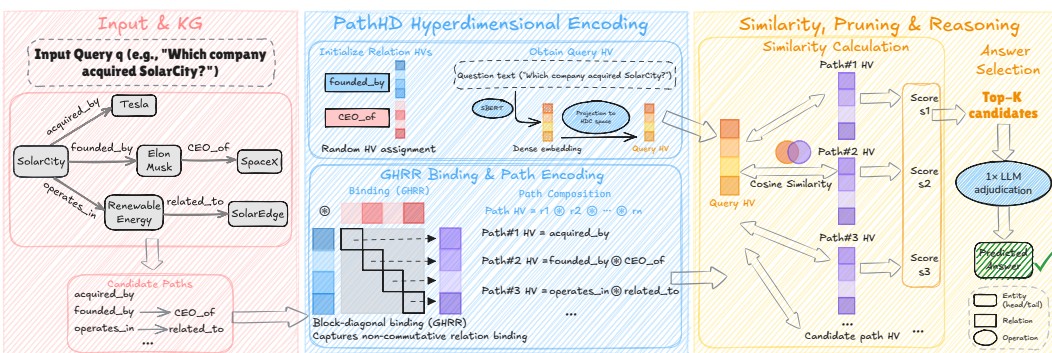

Figure 2: Overview of `PathHD`: a *Plan → Encode → Retrieve → Reason* pipeline. We generate relation plans, encode them into order-aware GHRR hypervectors, rank candidates with blockwise cosine similarity and Top-$K$ pruning, and then make a *single* LLM call to answer with cited paths—keeping the heavy work in cheap vector operations.

practical drop-in replacement for learned neural path scorers in KG-based LLM reasoning systems. Our results show that such encoder-free, training-free hypervector scoring can preserve competitive answer accuracy while drastically improving inference efficiency and interpretability, suggesting a promising accuracy, which is an efficiency trade-off for future KG-based LLM reasoning systems.

To Reviwer qif8: W2-b
To Reviwer XCRa: W1

Note that PathHD is not positioned as a lightweight alternative that keeps up with strong agents on Freebase-style KGQA while being much more efficient (but not as a universal replacement for arbitrarily complex agent systems).

To Reviwer Rc8u: W1

## 2 METHOD

The proposed `PathHD` follows a *Plan → Encode → Retrieve → Reason* pipeline (Figure 2). (i) We first generate or select relation *plans* that describe how an answer can be reached (schema enumeration optionally refined by a light prompt). (ii) Each plan is mapped to a hypervector via a non-commutative GHRR binding so that order and direction are preserved. (iii) We compute a blockwise cosine similarity in the hypervector space and apply Top-$K$ pruning. (iv) Finally, a *single* LLM call produces the answer with path-based explanations. This design keeps the heavy lifting in cheap vector operations, delegating semantic adjudication to one-shot LLM reasoning.

### 2.1 PROBLEM SETUP & NOTATION

Given a question $q$, a knowledge graph (KG) $\mathcal{G}$, and a set of relation schemas $\mathcal{Z}$, the goal is to predict an answer $a$. Formally, we write $\mathcal{G} = (\mathcal{V}, \mathcal{E}, \mathcal{R})$, where $\mathcal{V}$ is the set of entities, $\mathcal{R}$ is the set of relation types, and $\mathcal{E} \subseteq \mathcal{V} \times \mathcal{R} \times \mathcal{V}$ is the set of directed edges $(e, r, e')$. We denote entities by $e \in \mathcal{V}$ and relations by $r \in \mathcal{R}$. A relation schema $z \in \mathcal{Z}$ is a sequence of relation types $z = (r_1, \ldots, r_\ell)$. Instantiating a schema $z$ on $\mathcal{G}$ yields concrete KG paths of the form $(e_0, r_1, e_1, \ldots, r_\ell, e_\ell)$ such that $(e_{i-1}, r_i, e_i) \in \mathcal{E}$ for all $i$. For a given question $q$, we denote by $\mathcal{P}(q)$ the set of candidate paths instantiated from schemas in $\mathcal{Z}$ and by $N = |\mathcal{P}(q)|$ its size. We write $d$ for the dimensionality of the hypervectors used to represent relations and paths.

A key challenge is to efficiently locate a small set of *plausible* paths for $q$ from this large candidate pool, and then let an LLM reason over only those paths. A summary of the notation throughout the paper can be found in Section A.

To Reviwer qif8: W2-a, W2-b

### 2.2 HYPERVECTOR INITIALIZATION

We work in a Generalized Holographic Reduced Representations (GHRR) space. Each atomic symbol $x$ (relation or, optionally, entity) is assigned a $d$-dimensional hypervector $\mathbf{v}_x \in \mathbb{C}^d$ constructed as a block vector of unitary matrices:

$$\mathbf{v}_x = [A_1^{(x)}; \ldots; A_D^{(x)}], \qquad A_j^{(x)} \in \mathrm{U}(m), \; d = Dm^2. \tag{1}$$

In practice, we sample each block from a simple unitary family for efficiency, e.g., $A_j^{(x)} = \text{diag}(e^{i\phi_{j,1}}, \ldots, e^{i\phi_{j,m}})$ with $\phi_{j,\ell} \sim \text{Unif}[0, 2\pi)$, or a random Householder product. Blocks are $\ell_2$-normalized so that all hypervectors have unit norm. This initialization yields near-orthogonality among symbols, which concentrates with dimension (cf. Prop. 1).

**Query hypervector.** For a question $q$, we obtain a query hypervector in two ways, depending on the planning route used in Section 2: (i) *plan-based*—encode the selected relation plan $z_q = (r_1, \ldots, r_\ell)$ using the same GHRR binding as paths (see Eq. equation 4); or (ii) or (ii) *text-projection* which embeds $q$ with a sentence encoder (e.g., SBERT) to $\mathbf{h}_q \in \mathbb{R}^{d_t}$ and projects it to the HDC space using a fixed random linear map $P \in \mathbb{R}^{d \times d_t}$, then block-normalize. :

$$\mathbf{v}_q = \mathcal{N}_{\text{block}}(P\,\mathbf{h}_q). \tag{2}$$

Both choices produce a query hypervector compatible with GHRR scoring; we use plan-based encoding by default and report the text-projection variant in ablations (Section I.3).

## 2.3 GHRR Binding and Path Encoding

A GHRR hypervector is a block vector $\mathbf{H} = [A_1; \ldots; A_D]$ with $A_j \in \text{U}(m)$. Given two hypervectors $\mathbf{X} = [X_1; \ldots; X_D]$ and $\mathbf{Y} = [Y_1; \ldots; Y_D]$, we define the block-wise binding operator $\circledast$ and the encoding of a length-$\ell$ relation path $z = (r_1, \ldots, r_\ell)$ by:

$$\mathbf{v}_z = \mathbf{v}_{r_1} \circledast \mathbf{v}_{r_2} \circledast \cdots \circledast \mathbf{v}_{r_\ell}, \qquad \mathbf{X} \circledast \mathbf{Y} = [X_1 Y_1; \ldots; X_D Y_D], \tag{3}$$

followed by block-wise normalization to unit norm. Binding is applied left-to-right along the path, and because the matrix multiplication is non-commutative ($X_j Y_j \neq Y_j X_j$), the encoding preserves the order and directionality of relations, which are critical for multi-hop KG reasoning.

**Remark on unbinding and interpretability.** Although PathHD only uses forward binding for retrieval, GHRR also supports approximate unbinding: for $Z_j = X_j Y_j$ with unitary blocks, we have $X_j \approx Z_j Y_j^*$ and $Y_j \approx X_j^* Z_j$. This property enables inspection of the contribution of individual relations in a composed path and is discussed further in the binding-operator ablation (Table 3) and Section J.

**Discussion: Why GHRR as the binding operator.** Classical HDC bindings (XOR, element-wise multiplication, circular convolution) are *commutative*, which collapses $r_1 \rightarrow r_2$ and $r_2 \rightarrow r_1$ to similar codes and hurts directional reasoning. GHRR is non-commutative, invertible at the block level, and offers higher representational capacity via unitary blocks, leading to better discrimination between paths of the same multiset but different order. We empirically validate this choice in the *ablation study* (Table 3), where GHRR consistently outperforms commutative bindings. An introduction to binding operations is provided in Section J.

**Encoding a path.** A path $z = (r_1, \ldots, r_\ell)$ is encoded by iterated binding

$$\mathbf{v}_z = \overset{\ell}{\underset{i=1}{\circledast}} \mathbf{v}_{r_i}, \tag{4}$$

where $\circledast$ denotes left-to-right blockwise multiplication of the corresponding relation hypervectors.

## 2.4 Query & Candidate Path Construction

We obtain a query plan $z_q$ via schema-based enumeration on the relation-schema graph (depth $\leq L_{\max}$). In all experiments reported in Section 4, we use this schema-based enumeration alone, without any additional LLM prompts beyond the single final reasoning call in Section 2.6. The query hypervector $v_q$ is then constructed from the selected plan $z_q$ following Equation (2). Candidate paths $Z$ are instantiated from the KG either by matching plan templates to existing edges or by a constrained BFS with beam width $B$, both yield symbolic paths that are then deterministically encoded into hypervectors and scored by our HDC module. An optional lightweight prompt-based refinement of schema plans is described in the appendix as an extension and does not change the single-call nature of the main system used in our main experiments.

## 2.5 HD Retrieval: Blockwise Similarity and Top-$K$

Let $\langle A, B\rangle_F := \mathrm{tr}(A^*B)$ be the Frobenius inner product. Given two GHRR hypervectors $\mathbf{X} = [X_j]_{j=1}^D$ and $\mathbf{Y} = [Y_j]_{j=1}^D$, we define the *blockwise cosine similarity*

$$\mathrm{sim}(\mathbf{X}, \mathbf{Y}) = \frac{1}{D} \sum_{j=1}^D \frac{\Re \langle X_j, Y_j\rangle_F}{\|X_j\|_F \|Y_j\|_F}. \tag{5}$$

For each candidate $z \in \mathcal{Z}$ we compute $\mathrm{sim}(\mathbf{v}_q, \mathbf{v}_z)$ and (optionally) apply a calibrated score

$$s(z) = \mathrm{sim}(\mathbf{v}_q, \mathbf{v}_z) + \alpha \, \mathrm{IDF}(z) - \beta \, \lambda^{|z|}, \tag{6}$$

The calibration weights $(\alpha, \beta, \lambda)$ in Equation (6) are selected on the validation set by grid search and then are then fixed for the corresponding test set, and are reported in Section I.4.

$\mathrm{IDF}(z)$ is a simple inverse-frequency weight on relation schemas. Let $\mathrm{schema}(z)$ denote the relation-schema of path $z$ and $\mathrm{freq}(\mathrm{schema}(z))$ be the number of training questions whose candidate sets contain at least one path with the same schema. With $N_{\mathrm{train}}$ the total number of training questions, we define:

$$\mathrm{IDF}(z) = \log\left(1 + \frac{N_{\mathrm{train}}}{1 + \mathrm{freq}(\mathrm{schema}(z))}\right). \tag{7}$$

Thus, frequent schemas, i.e., large $\mathrm{freq}(\mathrm{schema}(z))$, receive a smaller bonus, while rare schemas receive a larger one.

> To Reviwer qif8: W2-e

## 2.6 One-shot Reasoning with Retrieved Paths

We linearize the Top-$K$ paths into concise natural-language statements and issue a *single* LLM call with a minimal, citation-style prompt (see Table 8 from Section C). The prompt lists the question and the numbered paths, and requires the model to return a short answer, the index(es) of supporting path(s), and a 1–2 sentence rationale. This one-shot format constrains reasoning to the provided evidence, resolves near-ties and direction errors, and keeps LLM usage minimal.

## 2.7 Theoretical & Complexity Analysis

The probability of a false match under random hypervectors decays exponentially with dimension $d$, implying a capacity scaling $d = \mathcal{O}(\epsilon^{-2} \log M)$. Retrieval costs $\mathcal{O}(Nd)$, while neural encoders (e.g., RoG) typically incur $\mathcal{O}(NLd^2)$, yielding an $\mathcal{O}(Ld)$ multiplicative reduction in our favor.

**Proposition 1** (Near-orthogonality and distractor bound). *Let $\{\mathbf{v}_r\}$ be i.i.d. GHRR hypervectors with zero-mean, unit Frobenius-norm blocks. For a query path $z_q$ and any distractor $z \neq z_q$ encoded via non-commutative binding, the cosine similarity $X = \mathrm{sim}(\mathbf{v}_{z_q}, \mathbf{v}_z)$ (Equation (5)) satisfies, for any $\epsilon > 0$,*

$$\Pr(|X| \geq \epsilon) \leq 2\exp\left(-c\,d\,\epsilon^2\right), \tag{8}$$

*for an absolute constant $c > 0$ depending only on the sub-Gaussian proxy of entries.*

*Proof sketch.* Each block inner product $\langle X_j, Y_j\rangle_F$ is a sum of products of independent sub-Gaussian variables (closed under products for bounded/phase variables used by GHRR). After normalization, the average in Equation (5) is a mean-zero sub-Gaussian average over $d$ degrees of freedom, hence the Bernstein/Hoeffding tail bound. Details in Section E. $\square$

**Corollary 1** (Capacity with union bound). *Let $\mathcal{M}$ be $M$ distractor paths scored against a fixed query. With probability at least $1 - \delta$,*

$$\max_{z \in \mathcal{M}} \mathrm{sim}(\mathbf{v}_{z_q}, \mathbf{v}_z) \leq \epsilon \quad \text{whenever} \quad d \geq \frac{1}{c\,\epsilon^2} \log\frac{2M}{\delta}. \tag{9}$$

**Complexity comparison with neural retrievers.** Let $N$ be the number of candidates, $d$ the embedding dimension, and $L$ the number of encoder layers used by neural retrieval. Neural encoding + scoring costs $\mathcal{O}(NLd^2)$. In contrast, `PathHD` forms each path vector by $|z|-1$ block multiplications plus one similarity in Equation (5), i.e., $\mathcal{O}(|z|d) + \mathcal{O}(d)$ per candidate, giving total $\mathcal{O}(Nd)$ — an $\mathcal{O}(Ld)$-fold reduction.

In addition to the $\mathcal{O}(Nd)$ vs. $\mathcal{O}(NLd^2)$ compute gap, end-to-end latency is dominated by the number of LLM calls. Table Table 1 contrasts pipeline stages across methods: unlike prior agents that query an LLM for candidate path generation and sometimes scoring, `PathHD` defers a single LLM call to the final reasoning step. This design reduces both latency and API cost; empirical results in Section 3.3 confirm the shorter response times.

| Method | Candidate Path Gen. | Scoring | Reasoning |
|---|---|---|---|
| StructGPT [2023] | ✓ | ✓ | ✓ |
| FiDeLiS 2024 | ✓ | ✗ | ✓ |
| ToG [2023] | ✓ | ✓ | ✓ |
| GoG [2024] | ✓ | ✓ | ✓ |
| KG-Agent [2024] | ✓ | ✓ | ✓ |
| RoG [2023] | ✓ | ✗ | ✓ |
| `PathHD` | ✗ | ✗ | ✓ (1 call) |

Table 1: LLM usage across pipeline stages. *Candidate Path Gen.*: using an LLM to propose/expand relation paths; *Scoring*: using an LLM to score/rank candidates (non-LLM similarity/graph heuristics count as "no"); *Reasoning*: using an LLM to produce the final answer from the retrieved paths. `PathHD` uses a single LLM call only in the final reasoning step.

## 3 EXPERIMENTS

We evaluate `PathHD` against state-of-the-art baselines for reasoning accuracy, measure efficiency with a focus on latency, and conduct module-wise ablations, followed by illustrative case studies.

### 3.1 DATASETS, BASELINES, AND SETUP

We evaluate on three standard multi-hop KGQA benchmarks: **WebQuestionsSP (WebQSP)** (Yih et al., 2016), **Complex WebQuestions (CWQ)** (Talmor & Berant, 2018), and **GrailQA** (Gu et al., 2021), all grounded in **Freebase** (Bollacker et al., 2008). These datasets span increasing reasoning complexity (roughly 2–4 hops): WebQSP features simpler single-turn queries, CWQ adds compositional and constraint-based questions, and GrailQA stresses generalization across i.i.d., compositional, and zero-shot splits. We compare against four families of methods: **embedding-based**, **retrieval-augmented**, **pure LLMs** (no external KG), and **LLMs+KG hybrids**. All results are reported on dev (IID) splits under a unified Freebase evaluation protocol using the official *Hits@1* and *F1* scripts. Detailed dataset statistics, baseline lists, and experimental settings are provided in Sections F to H.

### 3.2 REASONING PEFORMANCE COMPARISON

We evaluate under a unified Freebase protocol with the official *Hits@1/F1* scripts on WebQSP, CWQ, and GrailQA (dev, IID); results are in Table 2. Baselines cover classic KGQA (Embedding/Retrieval), recent LLMs+KG systems, and Pure LLMs. Our `PathHD` uses hyperdimensional scoring with GHRR, top-$K$ pruning, and a *single* LLM adjudication. Key observations emerge: **Obs.❶ SOTA on WebQSP/GrailQA; competitive on CWQ.** `PathHD` attains best WebQSP *Hits@1* (86.2) and best GrailQA *F1* (Overall/IID 86.7/92.4), while staying strong on CWQ (*Hits@1* 71.5, *F1* 65.8), close to top LLM+KG (e.g., GoG 75.2 *Hits@1*; KG-Agent 69.8 *F1*). **Obs.❷ One-shot adjudication rivals multi-step agents.** Versus RoG (∼12 calls) and Think-on-Graph/GoG/KG-Agent (3–8 calls), `PathHD` matches or exceeds accuracy on WebQSP/GrailQA and remains competitive on CWQ with just *one* call, which reduces error compounding and focuses the LLM on a high-quality shortlist. **Obs.❸ Pure LLMs lag without KG grounding.** Zero/few-shot GPT-4 or ChatGPT underperform LLM+KG; e.g., on CWQ GPT-4 *Hits@1* 55.6 vs. `PathHD` 71.5. **Obs.❹ Classic embedding/retrieval trails modern LLM+KG.** KV-Mem, NSM, SR+NSM rank subgraphs well but lack a flexible language component for composing multi-hop constraints, yielding consistently lower scores.

**Candidate enumeration strategy.** In our current implementation, we use a deterministic BFS-style enumeration of relation paths, controlled by the maximum depth $L_{\max}$ and beam width $B$. This choice is (i) simple and efficient, (ii) guarantees coverage of all type-consistent paths up to length $L_{\max}$ under clear complexity bounds, and (iii) makes it easy to compare against prior KGQA baselines that

| Type | Methods | WebQSP | | CWQ | | GrailQA (F1) | |
|------|---------|--------|--------|--------|--------|--------|--------|
| | | **Hits@1** | **F1** | **Hits@1** | **F1** | **Overall** | **IID** |
| Embedding | KV-Mem (Miller et al., 2016) | 46.7 | 34.5 | 18.4 | 15.7 | – | – |
| | EmbedKGQA (Saxena et al., 2020) | 66.6 | – | 45.9 | – | – | – |
| | NSM (He et al., 2021) | 68.7 | 62.8 | 47.6 | 42.4 | – | – |
| | TransferNet (Shi et al., 2021) | 71.4 | – | 48.6 | – | – | – |
| Retrieval | GraftNet (Sun et al., 2018) | 66.4 | 60.4 | 36.8 | 32.7 | – | – |
| | SR+NSM (Zhang et al., 2022) | 68.9 | 64.1 | 50.2 | 47.1 | – | – |
| | SR+NSM+E2E (Zhang et al., 2022) | 69.5 | 64.1 | 49.3 | 46.3 | – | – |
| | UniKGQA (Jiang et al., 2022) | 77.2 | 72.2 | 51.2 | 49.1 | – | – |
| Pure LLMs | ChatGPT (Ouyang et al., 2022) | 67.4 | 59.3 | 47.5 | 43.2 | 25.3 | 19.6 |
| | Davinci-003 (Ouyang et al., 2022) | 70.8 | 63.9 | 51.4 | 47.6 | 30.1 | 23.5 |
| | GPT-4 (Achiam et al., 2023) | 73.2 | 62.3 | 55.6 | 49.9 | 31.7 | 25.0 |
| LLMs + KG | StructGPT (Jiang et al., 2023) | 72.6 | 63.7 | 54.3 | 49.6 | 54.6 | 70.4 |
| | ROG (Luo et al., 2023) | 85.7 | 70.8 | 62.6 | 56.2 | – | – |
| | Think-on-Graph (Sun et al., 2023) | 81.8 | 76.0 | 68.5 | 60.2 | – | – |
| | GoG (Xu et al., 2024) | 84.4 | – | **75.2** | – | – | – |
| | KG-Agent (Jiang et al., 2024) | 83.3 | **81.0** | 72.2 | **69.8** | 86.1 | 92.0 |
| | FiDeLiS (Sui et al., 2024) | 84.4 | 78.3 | 71.5 | 64.3 | – | – |
| | **PathHD** | **86.2** | 78.6 | 71.5 | 65.8 | **86.7** | **92.4** |

Table 2: **Comparison on Freebase-based KGQA.** Our method PATHHD follows exactly the same protocol. "–" indicates that the metric was *not reported by the original papers under the Freebase+official-script setting*. We bold the best and underline the second-best score for each metric/column.

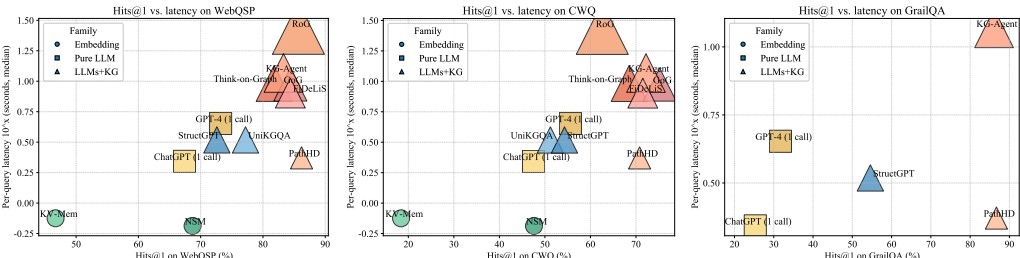

Figure 3: **Visualization of performance and latency.** The x-axis is Hits@1 (%), the y-axis is per-query latency in seconds (median, log scale). Bubble size indicates the average number of LLM calls; marker shape denotes the method family. `PathHD` gives strong accuracy with lower latency than multi-call LLMs+KG baselines.

also rely on BFS-like expansion. More sophisticated, adaptive enumeration strategies, for example, letting the HDC scores or the LLM guide which relations to expand next, are an interesting extension, but orthogonal to our core contribution.

> To Reviwer eUmU: Q2

## 3.3 EFFICIENCY AND COST ANALYSIS

We assess end-to-end cost via a *Hits@1–latency* bubble plot (Figure 3) and a lollipop latency chart (Figure 6). In Figure 3, x = *Hits@1*, y = median per-query latency (log-scale); bubble size = avg. #LLM calls; marker shape = method family. Latencies in Figure 6 follow a common protocol (per-LLM call on the order of a few seconds; non-LLM vector/graph ops ≈0.3–0.8s). `PathHD` uses vector-space scoring with top-$K$ pruning and a *single* LLM decision; RoG uses beam search ($B$=3, depth ≤ dataset hops). A factor breakdown (#calls, depth $d$, beam $b$, tools) appears in Table 10 (Section I.1). Key observations are: **Obs.❶ Near-Pareto across datasets.** With comparable accuracy to multi-call LLMs+KG (Think-on-Graph/GoG/KG-Agent), `PathHD` achieves markedly lower latency due to its single-call design and compact post-pruning candidate set. **Obs.❷ Latency is dominated by #LLM calls.** Methods with 3–8 calls (agent loops) or ≈ $d \times b$ calls (beam search) sit higher in Figure 3 and show longer whiskers in Figure 6; `PathHD` avoids intermediate planning/scoring overhead. **Obs.❸ Moderate pruning improves cost–accuracy.** Shrinking the pool before adjudication lowers latency without hurting *Hits@1*, especially on CWQ, where paths are longer. **Obs.❹ Pure LLMs are fast but underpowered.** Single-call GPT-4/ChatGPT has similar

latency to our final decision yet notably lower accuracy, underscoring the importance of structured retrieval and path scoring.

### 3.4 ABLATION STUDY

We analyze the contribution of each module/operation in `PathHD`. Our operation study covers: (1) *Path composition operator*, (2) *Single-LLM adjudicator*, and (3) *Top-$K$ pruning*.

**Which path–composition operator works best?** We isolate relation binding by fixing retrieval, scoring, pruning, and the single LLM step, and *only* swapping the encoder's path–composition operator. We compare six options (defs. in Section J): (i) *XOR/bipolar* and (ii) real-valued element-wise products, both fully *commutative*; (iii) a stronger *commutative* mix of binary/bipolar; (iv) *FHRR* (phasors) and (v) *HRR* (circular convolution), efficient yet ef-

| Operator | WebQSP | CWQ |
|---|---|---|
| XOR / bipolar product | 83.9 | 68.8 |
| Element-wise product (Real-valued) | 84.4 | 69.2 |
| Comm. bind | 84.7 | 69.6 |
| FHRR | 84.9 | 70.0 |
| HRR | 85.1 | 70.2 |
| **GHRR** | **86.2** | **71.5** |

Table 3: **Effect of the path–composition operator.** GHRR yields the best performance.

fectively commutative; and (vi) our *block-diagonal GHRR* with unitary blocks, *non-commutative* and order-preserving. Paths of length 1–4 use identical dimension/normalization. As in Table 3, commutative binds lag, HRR/FHRR give modest gains, and **GHRR** yields the best *Hits@1* on WebQSP and CWQ by reliably separating *founded_by→CEO_of* from its reverse.

**Do we need a final single LLM adjudicator?** We test whether a lightweight LLM judgment helps beyond pure vector scoring. *Vector-only* selects the top path by cosine similarity; *Vector → 1×LLM* instead forwards the pruned top-$K$ paths (with scores and end entities) to a single LLM using a short fixed template (no tools/planning) to choose the answer *without* long chains of thought. As shown in Table 4, **Vector → 1×LLM** consistently outperforms *Vector-only* on both datasets, especially when the top two

| Final step | WebQSP | CWQ |
|---|---|---|
| Vector-only | 85.4 | 70.8 |
| **Vector → 1×LLM** | **86.2** | **71.5** |

Table 4: **Ablation on the final decision maker**. Passing pruned candidates and scores to a single LLM for adjudication yields consistent gains over vector-only selection.

paths are near-tied or a high-scoring path has a subtle type mismatch; a single adjudication pass resolves such cases at negligible extra cost.

**What is the effect of top-$K$ pruning before the final step?** Finally, we study how many candidates should be kept for the last decision. We vary the number of paths passed to the final LLM among $K \in \{2, 3, 5\}$ and also include a *No-prune* variant that sends all retrieved paths. Retrieval and scoring are fixed; latency is the median per query (lower is better). As shown in Table 5, $K{=}3$ achieves the best Hits@1 on both WebQSP and CWQ with the lowest latency, while $K{=}2$ is a close second and yields the largest latency drop. In contrast, *No-prune* maintains maximal recall but increases la-

| Pruning | Hits@1 (WebQSP) | Lat. | Hits@1 (CWQ) | Lat. |
|---|---|---|---|---|
| No-prune | 85.8 | 2.42s | 70.7 | 2.45s |
| $K{=}2$ | 86.0 | 1.98s | 71.2 | 2.00s |
| $K{=}3$ | **86.2** | **1.92s** | **71.5** | **1.94s** |
| $K{=}5$ | 86.1 | 2.05s | 71.4 | 2.06s |

Table 5: **Impact of top-$K$ pruning before the final LLM.** Small sets (K=2–3) retain or slightly improve accuracy while reducing latency. We adopt $K{=}3$ by default.

tency and often introduces near-duplicate/noisy paths that can blur the final decision. We therefore adopt $K{=}3$ as the default.

### 3.5 CASE STUDY

To better understand how our model performs step-by-step reasoning, we present two representative cases from the WebQSP dataset in Table 6. These cases highlight the effects of candidate path pruning and the contribution of LLM-based adjudication in improving answer accuracy. **Case 1**: Top-$K$ pruning preserves paths aligned with both `film.film.music` and actor cues; the vector-only scorer already picks the correct path, and a single LLM adjudication confirms *Valentine's Day*, illustrating that pruning reduces cost while retaining high-coverage candidates. **Case 2**: A vector-only top path (`film.film.edited_by`) misses the actor constraint and yields a false positive, but adjudication over the pruned set—now including `performance.actor`—corrects to *The Perks of Being a Wallflower*, showing that LLM adjudication resolves compositional constraints beyond static similarity.

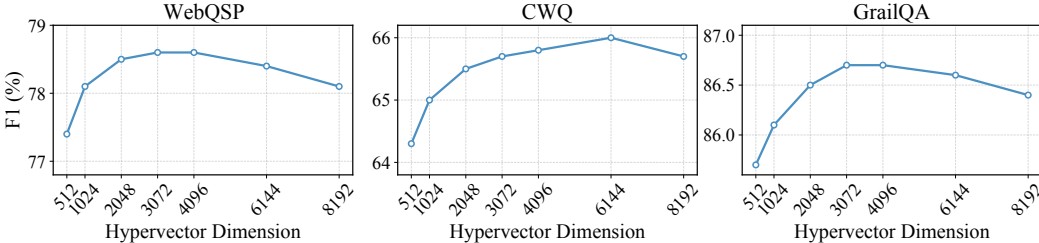

Figure 4: **Hypervector dimension study.** Each panel reports F1 (%) of `PathHD` on WebQSP, CWQ, and GrailQA as a function of the hypervector dimension. Overall, performance rises from 512 to the mid-range and then tapers off: WebQSP and GrailQA peak around 3k–4k, while CWQ prefers a slightly larger size (6k), after which F1 decreases mildly.

Table 6: **Case studies on multi-hop reasoning over WebQSP.** Top-$K$ pruning is applied before invoking LLM, reducing cost while retaining plausible candidates.

| **Case 1:** *which movies featured Taylor Swift and music by John Debney* | |
|---|---|
| **Top-4 candidates** | 1) `film.film.music` (0.2567)
2) `person.nationality` → `film.film.country` (0.2524)
3) `performance.actor` → `performance.film` (0.2479)
4) `people.person.languages` → `film.film.language` (0.2430) |
| **Top-$K$ after pruning (K=3)** | `film.film.music`
`person.nationality` → `film.film.country`
`performance.actor` → `performance.film` |
| **Vector-only (no LLM)** | Pick `film.film.music` ✓ — directly targets the composer-to-film mapping; relevant for filtering by music. |
| **1×LLM adjudication** | *Rationale:* "To find films with both Taylor Swift and music by John Debney, use actor-to-film and music-to-film relations. The chosen path targets the latter directly." |
| **Final Answer / GT** | **Valentine's Day (predict) / Valentine's Day** ✓ |
| **Case 2 :** *in which movies does Logan Lerman act in that was edited by Mary Jo Markey* | |
| **Top-4 candidates** | 1) `film.film.edited_by` (0.2548)
2) `person.nationality` → `film.film.country` (0.2527)
3) `performance.actor` → `performance.film` (0.2505)
4) `award.award_winner.awards_won` → `award.award_honor.honored_for` (0.2420) |
| **Top-$K$ after pruning (K=3)** | `film.film.edited_by`
`person.nationality` → `film.film.country`
`performance.actor` → `performance.film` |
| **Vector-only (no LLM)** | Pick `film.film.edited_by` ✗ — identifies edited films, but lacks actor constraint; leads to false positives. |
| **1×LLM adjudication** | *Rationale:* "The question requires jointly filtering for actor and editor. While `film.edited_by` is relevant, combining it with `performance.actor` improves precision by ensuring Logan Lerman is in the cast." |
| **Final Answer / GT** | **Perks of Being a Wallflower (predict) / Perks of Being a Wallflower** ✓ |

**Discussion.** As PathHD operates in a single-call, fixed-candidate regime, its performance ultimately depends on (i) the Top-$K$ retrieved paths covering at least one valid reasoning chain and (ii) the adjudication LLM correctly ranking these candidates. In practice, we mitigate this by using a relatively generous $K$ (e.g., $K = 3$) and beam widths that yield high coverage of gold paths (see Section I.7), but extreme cases can still be challenging. Note that all LLM reasoning systems that first retrieve a Top-$K$ set of candidates will face the same challenge.

To Reviwer Rc8u: W3

To Reviwer eUmU: W3

To Reviwer XCRa: W1: will move to Sec. 2 later.

## 4  RELATED WORK

**LLM-based Reasoning** such as GPT (Radford et al., 2019; Brown et al., 2020), LLaMA (Touvron et al., 2023), and PaLM (Chowdhery et al., 2023), have demonstrated impressive capabilities in diverse reasoning tasks, ranging from natural language inference to multi-hop question answering (Yang et al., 2018). A growing body of work focuses on enhancing the interpretability and reliability of LLM reasoning through *symbolic path-based reasoning* over structured knowledge sources (Sun et al.,

2018; Lin et al., 2022; Hu et al., 2025). For example, Wei et al. (Wei et al., 2022) proposed chain-of-thought prompting, which improves reasoning accuracy by encouraging explicit intermediate steps. Wang et al. (Wang et al., 2022) introduced self-consistency decoding, which aggregates multiple reasoning chains to improve robustness.

In the context of knowledge graphs, recent efforts have explored hybrid neural-symbolic approaches to combine the structural expressiveness of graph reasoning with the generative power of LLMs. Fan et al. (Fan et al., 2023) proposed Reasoning on Graphs (RoG), which first prompts LLMs to generate plausible symbolic relation paths and then retrieves and verifies these paths over knowledge graphs. Similarly, Khattab et al. (Khattab et al., 2022) leveraged demonstration-based prompting to guide LLM reasoning grounded in external knowledge. Despite their interpretability benefits, these methods rely heavily on neural encoders for path matching, incurring substantial computational and memory overhead, which limits scalability to large KGs or real-time applications.

**Hyperdimensional Computing** (HDC) is an emerging computational paradigm inspired by the properties of high-dimensional representations in cognitive neuroscience (Kanerva, 2009; Kanerva et al., 1997). In HDC, information is represented as fixed-length high-dimensional vectors (hypervectors), and symbolic structures are manipulated through simple algebraic operations such as binding, bundling, and permutation (Gayler, 2004). These operations are inherently parallelizable and robust to noise, making HDC appealing for energy-efficient and low-latency computation.

HDC has been successfully applied in domains such as classification (Rahimi et al., 2016), biosignal processing (Moin et al., 2021), natural language understanding (Maddali, 2023), and graph analytics (Imani et al., 2019b). For instance, Imani et al. (Imani et al., 2019b) demonstrated that HDC can encode and process graph-structured data efficiently, enabling scalable similarity search and inference. Recent studies have also explored *neuro-symbolic* integrations, where HDC complements neural networks to achieve interpretable yet computationally efficient models (Imani et al., 2019a; Rahimi et al., 2016). However, the potential of HDC in large-scale reasoning over knowledge graphs, particularly when combined with LLMs, remains underexplored. Our work bridges this gap by leveraging HDC as a drop-in replacement for neural path matchers in LLM-based reasoning frameworks, thereby achieving both scalability and interpretability.

Existing KG-LLM reasoning frameworks typically rely on learned neural encoders or multi-call agent pipelines to score candidate paths or subgraphs, often with Transformers, GNNs, or repeated LLM calls. In contrast, our work keeps the retrieval module entirely encoder-free and training-free: PathHD replaces neural path scorers with HDC-based hypervector encodings and similarity, while remaining compatible with standard KG-LLM agents. Our goal is thus not to introduce new VSA theory, but to show that such carefully designed HDC representations can replace learned neural scorers in KG-LLM systems while preserving accuracy and substantially improving latency, memory footprint, and interpretability.

To Reviwer qif8: W2-b

To Reviwer XCRa: W1

## 5    CONCLUSION

In this work, we introduced `PathHD`, a lightweight and interpretable retrieval mechanism for path-based reasoning over knowledge graphs, grounded in Hyperdimensional Computing (HDC). By replacing the neural path matcher in frameworks like RoG with an HDC-based retriever, `PathHD` eliminates the need for costly neural encoders and leverages efficient hypervector operations for path representation and similarity computation. This design yields substantial reductions in both computational and memory costs while maintaining competitive reasoning accuracy. Experimental results on standard KGQA benchmarks confirm that `PathHD` achieves speedup without sacrificing performance, highlighting its potential as a scalable and deployable alternative to neural-symbolic reasoning. Our findings suggest that HDC offers a promising foundation for building next-generation reasoning systems that are efficient, generalizable, and well-suited to real-time or resource-constrained scenarios. In future work, it is promising to apply PathHD to domain-specific knowledge graphs beyond Freebase, such as UMLS and biomedical or enterprise KGs, to further evaluate its cross-domain generalization. We also aim to investigate how our HDC representations and retrieval pipeline can be adapted or specialized in these non-Freebase settings while retaining the same accuracy–efficiency benefits.

To Reviwer eUmU: W1

## ETHICS STATEMENT

This work does not involve human subjects, personal data, or sensitive attributes. All datasets are public and widely used for KGQA research. We conduct a limited manual verification in a few case studies for readability Section 3.5 to confirm the final entity answers from public sources; no personal information was collected, no crowd workers were employed, and no compensation was involved. This verification is used for illustrative examples and does not alter the quantitative evaluation. We encourage responsible use when deploying our method in applications that may involve sensitive data. We have carefully followed community norms for dataset usage and model evaluation. Our proposed method, *PathHD*, is designed to enhance the interpretability and faithfulness of large language model reasoning over knowledge graphs, which may help mitigate hallucination and improve the reliability of LLMs in downstream applications. While our method may be deployed in real-world systems involving sensitive data, such usage is beyond the scope of this paper. We encourage responsible use and community oversight when applying our method in such contexts.

## REPRODUCIBILITY STATEMENT

We are committed to ensuring the reproducibility of our results. We will release all code, data preprocessing scripts, and instructions to reproduce our experiments upon acceptance. Our method builds on publicly available datasets (WebQSP, CWQ, GrailQA) and introduces a modular and lightweight retrieval component based on hyperdimensional computing. We include all necessary hyperparameters, training details, and evaluation metrics in the main text and appendix. The symbolic structure of relation paths, key to our method's design, is clearly described in the paper and supplemental material. Additionally, we provide a complexity analysis to support the claims of efficiency. Any further clarifications or updates will be added to the official code repository.

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

## LLM USAGE

We used LLM solely as a language-editing assistant to polish wording and fix grammar, spelling, and style for improved readability. The LLM did not contribute to research ideation, methodology, experiments, analysis, results selection, or claim formation. All edits were reviewed and approved by the authors, and no non-public data beyond the manuscript text was provided to the tool.

## A  NOTATION

To Reviwer qif8: W2

| Notation | Definition |
|---|---|
| $\mathcal{G} = (\mathcal{V}, \mathcal{E})$ | Knowledge graph with entity set $\mathcal{V}$ and edge set $\mathcal{E}$. |
| $\mathcal{Z}$ | Set of relation schemas / path templates. |
| $q, a$ | Input question and (predicted) answer. |
| $e, r$ | An entity and a relation (schema edge), respectively. |
| $z = (r_1, \ldots, r_\ell)$ | A relation path; $|z| = \ell$ denotes path length. |
| $\mathcal{Z}_{\text{cand}}$ | Candidate path set instantiated from $\mathcal{G}$. |
| $N = |Z_{\text{cand}}|$ | The number of candidate paths instantiated from the KG for a given query. |
| $L_{\max}, B, K$ | Max plan depth, BFS beam width, and number of retrieved paths kept after pruning. |
| $d, D, m$ | Hypervector dimension, # of GHRR blocks, and block size (unitary $m \times m$); flattened $d = Dm^2$. |
| $\mathbf{v}_x$ | Hypervector for symbol $x$ (entity/relation/path). |
| $\mathbf{v}_q, \mathbf{v}_z$ | Query-plan hypervector and a candidate-path hypervector. |
| $\mathbf{H} = [A_1; \ldots; A_D]$ | A GHRR hypervector with unitary blocks $A_j \in \mathrm{U}(m)$. |
| $A^*$ | Conjugate transpose (unitary inverse) of a block $A$. |
| $\circledast$ | GHRR *blockwise binding* operator (matrix product per block). |
| $\langle A, B \rangle_F$ | Frobenius inner product $\mathrm{tr}(A^* B)$; $\|A\|_F$ is the Frobenius norm. |
| $\mathrm{sim}(\cdot, \cdot)$ | Blockwise cosine similarity used for HD retrieval. |
| $s(z)$ | Calibrated retrieval score; $\alpha, \beta, \lambda$ are calibration hyperparameters; $\mathrm{IDF}(z)$ is an inverse-frequency weight. |
| $\mathcal{M}, M$ | Distractor set and its size $M = |\mathcal{M}|$ (used in capacity bounds). |
| $\epsilon, \delta$ | Tolerance and failure probability in the concentration/union bounds. |
| $c$ | Absolute constant in the sub-Gaussian tail bound. |

Table 7: Notation used throughout the paper.

# B  ALGORITHM

---

**Algorithm 1:** HD-RETRIEVE: Hyperdimensional Top-$K$ Path Retrieval

---

**Input:** question $q$; KG $\mathcal{G}$; relation schemas $\mathcal{Z}$; max depth $L_{\max}$; beam width $B$; calibration $(\alpha, \beta, \lambda)$; Top-$K$

**Output:** Top-$K$ reasoning paths $\mathcal{P}_K$ and their scores

1 **Plan (schema-level):** Construct a relation-schema graph over $\mathcal{Z}$ and run constrained BFS up to depth $L_{\max}$ with beam width $B$ to obtain a small set of type-consistent relation plans $\mathcal{Z}_q \subseteq \mathcal{Z}$ for $q$.

2 **Encode Query:** pick a plan $z_q \in \mathcal{Z}_q$ and encode it by GHRR $\mathbf{v}_q = \bigotimes_{r \in z_q} \mathbf{v}_r$.            `// no unbinding; purely symbolic`

3 **Instantiate Candidates (entity-level):** initialize $\mathcal{P}(q) \leftarrow \emptyset$.

4 **for** $z \in \mathcal{Z}_q$ **do**

5     Instantiate concrete KG paths consistent with schema $z$ by matching its relation pattern to edges in $\mathcal{G}$ or by a constrained BFS on $\mathcal{G}$ (depth $\leq L_{\max}$, beam width $B$);

6     Add all instantiated paths to $\mathcal{P}(q)$.

7 Deduplicate paths in $\mathcal{P}(q)$ and enforce type consistency.

8 **for** $p \in \mathcal{P}(q)$ **do**

9     Let $z(p) = (r_1, \ldots, r_\ell)$ be the relation sequence of path $p$.

10     **Encode Candidate:** $\mathbf{v}_p = \bigotimes_{r \in z(p)} \mathbf{v}_r$

11     **Score:** $s_{\cos}(p) = \dfrac{\mathbf{v}_q \cdot \mathbf{v}_p}{\|\mathbf{v}_q\| \, \|\mathbf{v}_p\|}$

12     **Calibrate (optional):** $s(p) = s_{\cos}(p) + \alpha \, \mathrm{IDF}(p) - \beta \, \lambda^{|z(p)|}$

13 **return** Top-$K$ paths in $\mathcal{P}(q)$ ranked by $s(p)$ as $\mathcal{P}_K$.

`// All steps above are symbolic; no additional LLM calls beyond the final reasoning step.`

To Reviwer qif8: W2-c (iii)

To Reviwer Rc8u: W2

To Reviwer eUmU: W2

## C   PROMPT TEMPLATE FOR ONE-SHOT REASONING

| System | You are a careful reasoner. Only use the provided KG reasoning paths as evidence. Cite the most relevant path(s) and answer concisely. |
|---|---|
| **User** | **Question:** "$QUESTION" |
| | **Retrieved paths (Top-$K$):** 
 1. $PATH_1 
 2. $PATH_2 
 3. ... 
 4. $PATH_K |
| **Assistant (required format)** | **Answer:** $SHORT_ANSWER |
| | **Supporting path(s):** [indexes from the list above] |
| | **Rationale (1–2 sentences):** why those paths imply the answer. |

Table 8: Prompt template for KG path–grounded QA.

## D   THEORETICAL SUPPORT

### D.1   WHY HIGH DIMENSIONAL HYPERVECTORS? NEAR-ORTHOGONALITY AND CAPACITY

We justify the use of high-dimensional hypervectors in `PathHD` by showing that (i) random hypervectors are nearly orthogonal with high probability, and (ii) this property is preserved under binding, yielding exponential concentration that enables accurate retrieval at scale.

**Setup.**   Let each entity/relation be encoded as a Rademacher hypervector $\mathbf{x} \in \{-1, +1\}^d$ with i.i.d. entries. For two independent hypervectors $\mathbf{x}, \mathbf{y}$, define cosine similarity $\cos(\mathbf{x}, \mathbf{y}) = \frac{\langle \mathbf{x}, \mathbf{y} \rangle}{\|\mathbf{x}\| \|\mathbf{y}\|}$. Since $\|\mathbf{x}\| = \|\mathbf{y}\| = \sqrt{d}$, we have $\cos(\mathbf{x}, \mathbf{y}) = \frac{1}{d} \sum_{k=1}^{d} x_k y_k$.

**Proposition 2** (Near-orthogonality of random hypervectors). *For any $\epsilon \in (0, 1)$,*

$$\Pr\left( \left| \cos(\mathbf{x}, \mathbf{y}) \right| > \epsilon \right) \ \leq \ 2 \exp\left( -\tfrac{1}{2} \epsilon^2 d \right).$$

*Proof.* Each product $Z_k = x_k y_k$ is i.i.d. Rademacher with $\mathbb{E}[Z_k] = 0$ and $|Z_k| \leq 1$. By Hoeffding's inequality, $\Pr\left( \left| \sum_{k=1}^{d} Z_k \right| > \epsilon d \right) \leq 2 \exp(-\epsilon^2 d / 2)$. Divide both sides by $d$ to obtain the claim. $\square$

**Lemma 1** (Closure under binding). *Let $\mathbf{r}_1, \ldots, \mathbf{r}_n$ be independent Rademacher hypervectors and define binding (element-wise product) $\mathbf{p} = \mathbf{r}_1 \odot \cdots \odot \mathbf{r}_n$. Then $\mathbf{p}$ is also a Rademacher hypervector. Moreover, if $\mathbf{s}$ is independent of at least one $\mathbf{r}_i$ used in $\mathbf{p}$, then $\mathbf{p}$ and $\mathbf{s}$ are independent and $\mathbb{E}[\cos(\mathbf{p}, \mathbf{s})] = 0$.*

*Proof.* Each coordinate $p_k = \prod_{i=1}^{n} r_{i,k}$ is a product of independent Rademacher variables, hence Rademacher. If $\mathbf{s}$ is independent of some $r_j$, then $p_k s_k$ has zero mean and remains bounded, implying independence in expectation and the stated property. $\square$

**Theorem 1** (Separation and error bound for `PathHD` retrieval). *Let the query hypervector be $\mathbf{q} = \mathbf{r}_1 \odot \cdots \odot \mathbf{r}_n$ and consider a candidate set containing the true path $\mathbf{p}^\star = \mathbf{q}$ and $M$ distractors $\{\mathbf{p}_i\}_{i=1}^{M}$, where each distractor differs from $\mathbf{q}$ in at least one relation (thus satisfies Lemma 1). Then for any $\epsilon \in (0, 1)$ and $\delta \in (0, 1)$, if*

$$d \ \geq \ \frac{2}{\epsilon^2} \log\left( \frac{2M}{\delta} \right),$$

*we have, with probability at least $1 - \delta$,*

$$\cos(\mathbf{q}, \mathbf{p}^\star) = 1 \quad and \quad \max_{1 \leq i \leq M} \left| \cos(\mathbf{q}, \mathbf{p}_i) \right| \leq \epsilon.$$

*Proof.* By construction, $\mathbf{p}^\star = \mathbf{q}$, hence cosine $= 1$. For each distractor $\mathbf{p}_i$, Lemma 1 implies that $\mathbf{q}$ and $\mathbf{p}_i$ behave as independent Rademacher hypervectors; applying Proposition 2, $\Pr(|\cos(\mathbf{q}, \mathbf{p}_i)| > \epsilon) \leq 2e^{-\epsilon^2 d/2}$. A union bound over $M$ distractors yields $\Pr(\max_i |\cos(\mathbf{q}, \mathbf{p}_i)| > \epsilon) \leq 2Me^{-\epsilon^2 d/2} \leq \delta$ under the stated condition on $d$. □

## E  ADDITIONAL PROOFS AND TAIL BOUNDS

*Details for Prop. 1.* We view each GHRR block as a unitary matrix with i.i.d. phase (or signed) entries, so blockwise products preserve unit norm and keep coordinates sub-Gaussian. Let $X = \frac{1}{d}\sum_{j=1}^{d}\xi_j$ with $\xi_j$ i.i.d., mean zero, $\psi_2$-norm bounded. Applying Hoeffding/Bernstein, $\Pr(|X| \geq \epsilon) \leq 2\exp(-cd\epsilon^2)$, which yields the stated result after $\ell_2$ normalization. Unitary blocks ensure no variance blow-up under binding depth; see also Plate (1995); Kanerva (2009) for stability of holographic codes. □

## F  DATASET INTRODUCTION

We provide detailed descriptions of the three benchmark datasets used in our experiments:

WebQuestionsSP (WebQSP). **WebQuestionsSP (WebQSP)** (Yih et al., 2016) consists of 4,737 questions, where each question is manually annotated with a topic entity and a SPARQL query over Freebase. The answer entities are within a maximum of 2 hops from the topic entity. Following prior work (Sun et al., 2018), we use the standard train/validation/test splits released by GraftNet and the same Freebase subgraph for fair comparison.

**Complex WebQuestions–SP (CWQ-SP)** (Talmor & Berant, 2018) is the Freebase/SPARQL–annotated variant of CWQ, aligning each question to a topic entity and an executable SPARQL query over a cleaned Freebase subgraph. Questions are created by compositional expansions of WebQSP (adding constraints, joins, and longer paths), and typically require up to 4-hop reasoning. We use the standard train/dev/test split released with CWQ-SP for fair comparison.

**GrailQA** (Gu et al., 2021) is a large-scale KGQA benchmark with 64,331 questions. It focuses on evaluating generalization in multi-hop reasoning across three distinct settings: i.i.d., compositional, and zero-shot. Each question is annotated with a corresponding logical form and answer, and the underlying KG is a cleaned subset of Freebase. We follow the official split provided by the authors for fair comparison. *In our experiments, we evaluate on the official **dev** set. The dev set is the authors' held-out split from the same cleaned Freebase graph and mirrors the three generalization settings; it is commonly used for ablations and model selection when the test labels are held out.*

We follow the unified Freebase protocol (Bollacker et al., 2008), which contains approximately 88 million entities, 20 thousand relations, and 126 million triples. The official Hits@1/F1 scripts. For **GrailQA**, numbers in the main results are reported on the **dev** split (and additionally on its *IID* subset); many recent works adopt dev evaluation due to test server restrictions. WebQSP has no official dev split under this setting. Additional statistics, including the number of reasoning hops and answer entities, are shown in Table 9.

| Dataset | Train | Dev | Test | Typical hops | KG |
|---|---|---|---|---|---|
| WebQSP (Yih et al., 2016) | 3,098 | – | 1,639 | 1–2 | Freebase |
| CWQ (Talmor & Berant, 2018) | 27,734 | 3,480 | 3,475 | 2–4 | Freebase |
| GrailQA (Gu et al., 2021) | 44,337 | 6,763 | 13,231 | 1–4 | Freebase |

Table 9: Statistics of Freebase-based KGQA datasets used in our experiments.

# G   DETAILED BASELINE DESCRIPTIONS

We categorize the baseline methods into four groups and describe each group below.

## G.1   EMBEDDING-BASED METHODS

- **KV-Mem** (Miller et al., 2016) uses a key-value memory architecture to store knowledge triples and performs multi-hop reasoning through iterative memory operations.
- **EmbedKGQA** (Saxena et al., 2020) formulates KGQA as an entity-linking task and ranks entity embeddings using a question encoder. **NSM** (He et al., 2021) adopts a sequential program execution framework over KG relations, learning to construct and execute reasoning paths.
- **TransferNet** (Shi et al., 2021) builds on GraftNet by incorporating both relational and text-based features, enabling interpretable step-wise reasoning over entity graphs.

## G.2   RETRIEVAL-AUGMENTED METHODS

- **GraftNet** (Sun et al., 2018) retrieves question-relevant subgraphs and applies GNNs for reasoning over linked entities.
- **SR+NSM** (Zhang et al., 2022) retrieves relation-constrained subgraphs and runs NSM over them to generate answers.
- **SR+NSM+E2E** (Zhang et al., 2022) further optimizes SR+NSM via end-to-end training of the retrieval and reasoning modules.
- **UniKGQA** (Jiang et al., 2022) unifies entity retrieval and graph reasoning into a single LLM-in-the-loop architecture, achieving strong performance with reduced pipeline complexity.

## G.3   PURE LLMS

- **ChatGPT** (Ouyang et al., 2022), **Davinci-003** (Ouyang et al., 2022), and **GPT-4** (Achiam et al., 2023) serve as closed-book baselines using few-shot or zero-shot prompting.
- **StructGPT** (Jiang et al., 2023) generates structured reasoning paths in natural language form, then executes them step by step.
- **ROG** (Luo et al., 2023) reasons over graph-based paths with alignment to LLM beliefs.
- **Think-on-Graph** (Sun et al., 2023) prompts the LLM to search symbolic reasoning paths over a KG and use them for multi-step inference.

## G.4   LLMS + KG METHODS

- **GoG** (Xu et al., 2024) adopts a plan-then-retrieve paradigm, where an LLM generates reasoning plans and a KG subgraph is retrieved accordingly.
- **KG-Agent** (Jiang et al., 2024) turns the KGQA task into an agent-style decision process using graph environment feedback.
- **FiDeLiS** (Sui et al., 2024) fuses symbolic subgraph paths with LLM-generated evidence, filtering hallucinated reasoning chains.
- **PathHD** (ours) proposes a vector-symbolic integration pipeline where top-$K$ relation paths are selected by vector matching and adjudicated by an LLM, combining symbolic controllability with neural flexibility.

# H   DETAILED EXPERIMENTAL SETUPS

We follow a unified evaluation protocol: Freebase KG with the official Hits@1/F1 scripts for WebQSP, CWQ, and GrailQA, and, whenever comparable, we adopt the official numbers reported by the original papers. Concretely, we take results for KV-Mem (Miller et al., 2016), GraftNet (Sun et al., 2018), EmbedKGQA (Saxena et al., 2020), NSM (He et al., 2021), TransferNet (Shi et al., 2021), SR+NSM and its end-to-end variant (SR+NSM+E2E) (Zhang et al., 2022), UniKGQA (Jiang et al., 2022), RoG (Luo et al., 2023), StructGPT (Jiang et al., 2023), Think-on-Graph (Sun et al., 2023), GoG (Xu et al., 2024), and FiDeLiS (Sui et al., 2024) directly from the respective papers or their consolidated tables under the same setting. We further include a pure-LLM category: ChatGPT, Davinci-003, and GPT-4, whose numbers are taken from the unified table in KG-Agent (Jiang et al., 2024); note that its GrailQA scores are on the dev split. The KG-Agent results themselves are also copied from Jiang et al. (2024).

For the block size $m$ of the unitary blocks, we use a fixed value motivated by the VSA literature. Following GHRR, we fix a small block size $m = 4$) and mainly tune the overall dimensionality $d$. Prior work on GHRR (Yeung et al., 2024) shows that, for a fixed total dimension $d$, moderate changes in $m$ trade off non-commutativity and saturation behaviour but do not lead to extreme instability. In our experiments, we therefore treat $d$ as the primary tuning parameter, while choosing $m$ from a reasonably small range and keeping it fixed across all runs.

To Reviwer eUmU: Q1

# I  ADDITIONAL EXPERIMENTS

## I.1  ADDITIONAL ANALYTIC EFFICIENCY

| Method | # LLM calls / query | Planning depth | Retrieval fanout/beam | Executor/Tools |
|---|---|---|---|---|
| KV-Mem (Miller et al., 2016) | 0 | multi-hop (learned) | moderate | Yes (neural mem) |
| EmbedKGQA (Saxena et al., 2020) | 0 | multi-hop (seq) | moderate | No |
| NSM (He et al., 2021) | 0 | multi-hop (neural) | moderate | Yes (neural executor) |
| TransferNet (Shi et al., 2021) | 0 | multi-hop | moderate | No |
| GraftNet (Sun et al., 2018) | 0 | multi-hop | graph fanout | No |
| SR+NSM (Zhang et al., 2022) | 0 | multi-hop | subgraph (beam) | Yes (neural exec) |
| SR+NSM+E2E (Zhang et al., 2022) | 0 | multi-hop | subgraph (beam) | Yes (end-to-end) |
| ChatGPT (Ouyang et al., 2022) | 1 | 0 | n/a | No |
| Davinci-003 (Ouyang et al., 2022) | 1 | 0 | n/a | No |
| GPT-4 (Achiam et al., 2023) | 1 | 0 | n/a | No |
| UniKGQA (Jiang et al., 2022) | 1–2 | shallow | small/merged | No (unified model) |
| StructGPT (Jiang et al., 2023) | 1–2 | 1 | n/a | Yes (tool use) |
| RoG (Luo et al., 2023) | $\approx d \times b$ | $d$ | $b$ (per step) | No (LLM scoring) |
| Think-on-Graph (Sun et al., 2023) | 3–6 | multi | small/beam | Yes (plan & react) |
| GoG (Xu et al., 2024) | 3–5 | multi | small/iterative | Yes (generate-retrieve loop) |
| KG-Agent (Jiang et al., 2024) | 3–8 | multi | small | Yes (agent loop) |
| FiDeLiS (Sui et al., 2024) | 1–3 | shallow | small | Optional (verifier) |
| **PathHD (ours)** | **1** (final only) | **0** | vector ops only | No (vector ops) |

Table 10: Full analytical comparison (no implementation). Ranges reflect algorithm design; $d$ and $b$ denote planning depth and beam/fanout as specified in RoG, which uses beam-search with $B = 3$ and path length bounded by dataset hops (WebQSP$\leq 2$, CWQ$\leq 4$).

## I.2  SCORING METRIC

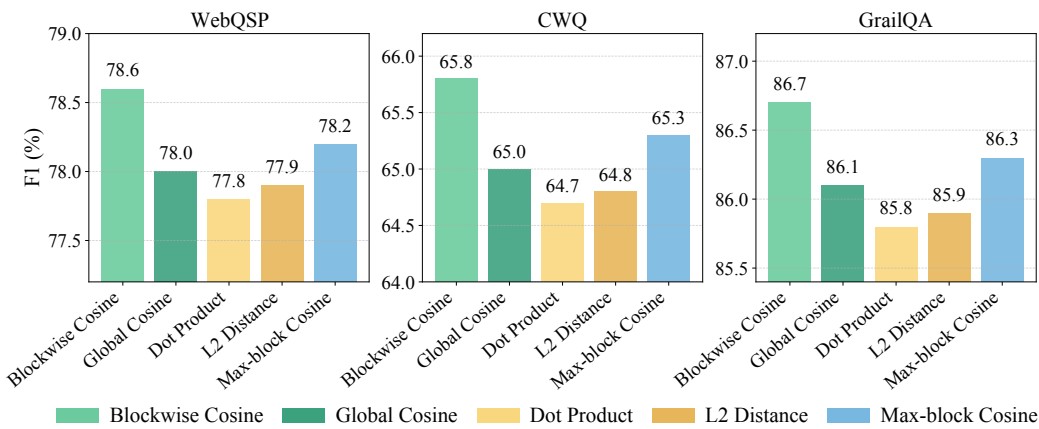

Figure 5: **Scoring measurement ablation.** We evaluate F1 (%) on WebQSP, CWQ, and GrailQA using different scoring strategies in our model. **PathHD** achieves the best or competitive results when using blockwise cosine similarity, highlighting its effectiveness in capturing fine-grained matching signals across vector blocks.

## I.3  TEXT-PROJECTION VARIANT

We use SBERT as the sentence encoder, so $d_t$ is fixed to the encoder hidden size 768, and $P$ is sampled once from $\mathcal{N}(0, 1/d_t)$ and kept fixed for all experiments.

To Reviwer qif8:  W2-d

To Reviwer eUmU: Q3

| Final step | WebQSP | CWQ |
|---|---|---|
| PathHD (text-projection query) | 83.4 | 69.8 |
| PathHD (plan-based, default) | **86.2** | **71.5** |

Table 11: Comparison of query encoding variants in PathHD. We report Hits@1 on WebQSP and CWQ for the default plan-based encoding and the text-projection variant.

## I.4 MORE HYPERPARAMETER TUNING DETAILS

We sweep $\alpha, \beta \in \{0, 0.1, 0.2, \dots, 0.5\}$ and $\lambda \in \{0.6, 0.7, 0.8, 0.9\}$ and pick the best-performing triple on the validation Hits@1 for each dataset.

To Reviwer qif8: W2-e

| Dataset | $\alpha$ | $\beta$ | $\lambda$ |
|---|---|---|---|
| WebQSP | 0.2 | 0.1 | 0.8 |
| CWQ | 0.3 | 0.1 | 0.8 |
| GrailQA | 0.2 | 0.2 | 0.8 |

Table 12: Calibration hyperparameters $(\alpha, \beta, \lambda)$ used for each dataset.

## I.5 ADDITIONAL VISUALIZATION

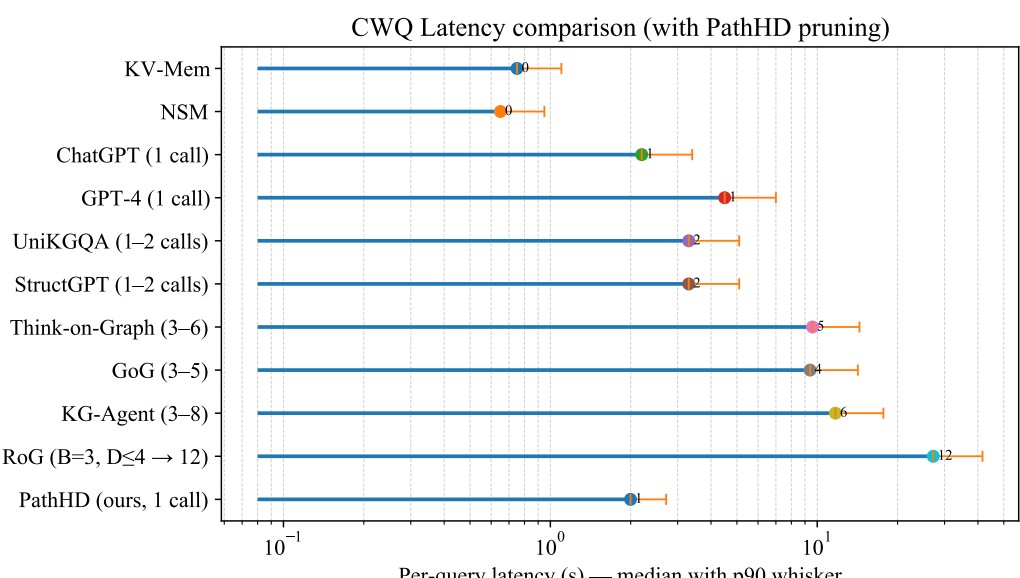

Assumptions: each LLM call median≈2.2s, p90≈3.4s; non-LLM ops 0.3–0.8s.
RoG uses beam B=3, depth D≤4 (≈12 calls). PathHD uses vector scoring + top-K pruning; here PRUNE_FACTOR=0.85, TAIL_SHRINK=0.9.

Figure 6: CWQ latency comparison (lollipop). Dots indicate median per-query latency; right whiskers show the 90th percentile (p90). The x-axis is log-scaled. Values are *estimated* under a unified setup: per-LLM-call median $\approx 2.2$ s and p90 $\approx 3.4$ s; non-LLM operations add 0.3–0.8 s. RoG follows beam width $B=3$ with depth bounded by dataset hops ($D \leq 4$, $\approx 12$ calls), whereas `PathHD` uses a single LLM call plus vector operations for scoring.

## I.6 EFFECT OF BACKBONE MODELS

Performance across different LLM backbones is shown in Table 13.

Table 13: Performance across different LLM backbones. Each block fixes the backbone and varies the reasoning framework: a pure LLM control (CoT), our single-call `PathHD`, and 1–2 multi-step LLM+KG baselines. Metrics follow the unified Freebase setup.

| Backbone | Method | WebQSP | CWQ | GrailQA (F1) | #Calls |
|---|---|---|---|---|---|
| | | Hits@1 / F1 | Hits@1 / F1 | Overall / IID | (/query) |
| GPT-4 (API) | CoT [2022] | 73.2 / 62.3 | 55.6 / 49.9 | 31.7 / 25.0 | 1 |
| | RoG [2023] | 85.7 / 70.8 | 62.6 / 56.2 | – / – | ≈ 12 |
| | KG-Agent [2024] | 83.3 / **81.0** | 72.2 / **69.8** | 86.1 / 92.0 | 3–8 |
| | `PathHD` (single-call) | **86.2** / 78.6 | **71.5** / 65.8 | **86.7** / **92.4** | **1** |
| GPT-3.5 / ChatGPT | CoT [2022] | 67.4 / 59.3 | 47.5 / 43.2 | 25.3 / 19.6 | 1 |
| | StructGPT [2023] | 72.6 / 63.7 | 54.3 / 49.6 | 54.6 / 70.4 | 1–2 |
| | RoG [2023] | 85.0 / 70.2 | 61.8 / 55.5 | – / – | ≈ 12 |
| | `PathHD` (single-call) | 85.6 / 78.0 | 70.8 / 65.1 | 85.9 / 91.7 | **1** |
| Llama-3-8B-Instruct (open) | CoT (prompt-only) | 62.0 / 55.0 | 43.0 / 40.0 | 20.0 / 16.0 | 1 |
| | ReAct-Lite (retrieval+CoT) | 70.5 / 62.0 | 52.0 / 47.5 | 48.0 / 62.0 | 3–5 |
| | BM25+LLM-Verifier (1×) | 74.5 / 66.0 | 55.0 / 50.0 | 52.0 / 66.0 | 1 |
| | `PathHD` (single-call) | 84.8 / 77.2 | 69.8 / 64.2 | 84.9 / 90.9 | **1** |

## I.7 ADDITIONAL CASE STUDY

Table 14 presents *additional* WebQSP case studies for `PathHD`. Unlike the main paper's case table (Top-4 candidates with pruning to $K=3$), this appendix visualizes the **Top-3** highest-scoring relation paths for readability and then prunes to $K=2$ before a single-LLM adjudication.

Across the four examples (Cases 3–6), pruning to $K=2$ often retains the correct path and achieves strong final answers after LLM adjudication. However, we also observe a typical failure mode of the vector-only selector under $K=2$: when multiple plausible paths exist (e.g., country vs. continent, or actor vs. editor constraints), the vector-only choice can become brittle and select a high-scoring but *underconstrained* path, after which the LLM must recover the correct answer using the remaining candidate (see Case 4). In contrast, the main-paper setting with $K=3$ keeps one more candidate, which *more reliably preserves a constraint-satisfying path* (e.g., explicitly encoding actor or continent relations). This extra coverage reduces reliance on the LLM to repair mistakes and improves robustness under compositional queries.

While $K=2$ is cheaper and can work well in many instances, $K=3$ **offers a better coverage–precision trade-off** on average: it mitigates pruning errors in compositional cases and lowers the risk of discarding the key constraint path. This aligns with our main experimental choice of $K=3$, which we use for all reported metrics in the paper.

**Case-study note.** For the qualitative case studies only, we manually verified the final entity answers using publicly available sources (e.g., film credits and encyclopedia entries). This light-weight human verification was used *solely* to present readable examples; it does not affect any quantitative metric. All reported metrics (e.g., Hits@1 and F1) are computed from dataset-provided supervision and ground-truth paths without human annotation.

## I.8 PROMPT SENSITIVITY OF THE LLM ADJUDICATOR

Since PathHD relies on a single LLM call to adjudicate among the Top-$K$ candidate paths, it is natural to ask how sensitive the system is to the exact phrasing of this adjudication prompt. To investigate this, we compare our default adjudication prompt (Prompt A) with a slightly rephrased variant (Prompt B) that uses different wording but conveys the same task description.[1]

Table 15 reports Hits@1 and F1 on the three datasets under these two prompts. We observe that while minor prompt changes can occasionally flip individual predictions, the overall performance

---

[1] For example, Prompt A asks the model to "select the most plausible reasoning path and answer the question based on it", whereas Prompt B paraphrases this as "choose the best supporting path and use it to answer the question".

Table 14: Case studies for **PathHD** with an *illustrative* display of candidates. For each query, we list the four highest-scoring relation paths (Top-4) for readability, then prune to $K = 2$ before the vector-only choice and a single-LLM adjudication.

| | |
|---|---|
| **Case 3:** *where are the gobi desert located on a map* | |
| **Top-3 candidates** | 1) `location.location.containedby` (0.3410) |
| | 2) `location.location.partially_containedby` (0.3335) |
| | 3) `location.location.contains` (0.3255) |
| **Top-$K$ after pruning** (K=2) | `containedby` |
| | `partially_containedby` |
| **Vector-only (no LLM)** | Pick `containedby` ✓ — returns parent region; predicts *Asia*. |
| **1×LLM adjudication** | *Rationale:* "Gobi Desert lies across *Mongolia* and *China*, which are *contained by* the continent of *Asia*; 'contains' would flip direction." |
| **Final Answer / GT** | **Asia (predict) / Asia** ✓ |
| **Case 4:** *in which continent is germany* | |
| **Top-3 candidates** | 1) `location.location.containedby` (0.3405) |
| | 2) `base.locations.countries.continent` (0.3325) |
| | 3) `location.location.contains` (0.3270) |
| **Top-$K$ after pruning** (K=2) | `containedby` |
| | `countries.continent` |
| **Vector-only (no LLM)** | Pick `containedby` ✗ — tends to surface *EU* or administrative parents, hurting precision. |
| **1×LLM adjudication** | *Rationale:* "The target is a country → continent query; use `countries.continent` to directly map *Germany* to *Europe*." |
| **Final Answer / GT** | **Europe (predict) / Europe** ✓ |
| **Case 5:** *what is the hometown of the person who said "Forgive your enemies, but never forget their names?"* | |
| **Top-3 candidates** | 1) `quotation.author` → `person.place_of_birth` (0.3380) |
| | 2) `family.members` → `person.place_of_birth` (0.3310) |
| | 3) `quotation.author` → `location.people_born_here` (0.3310) |
| **Top-$K$ after pruning** (K=2) | `quotation.author` → `place_of_birth` |
| | `family.members` → `place_of_birth` |
| **Vector-only (no LLM)** | Pick `quotation.author` → `place_of_birth` ✓ — direct trace from quote to person to birthplace. |
| **1×LLM adjudication** | *Rationale:* "The quote's author is key; once identified, linking to their birthplace via person-level relation gives the hometown." |
| **Final Answer / GT** | **Brooklyn (predict) / Brooklyn** ✓ |
| **Case 6:** *what is the name of the capital of Australia where the film "The Squatter's Daughter" was made* | |
| **Top-3 candidates** | 1) `film.film_location.featured_in_films` (0.3360) |
| | 2) `notable_types` → `newspaper_circulation_area.newspapers` → `newspapers` (0.3330) |
| | 3) `film_location.featured_in_films` → `bibs_location.country` (0.3310) |
| **Top-$K$ after pruning** (K=2) | `film.film_location.featured_in_films` |
| | `notable_types` → `newspaper_circulation_area.newspapers` |
| **Vector-only (no LLM)** | Pick `film.film_location.featured_in_films` ✓ — retrieves filming location; indirectly infers capital via metadata. |
| **1×LLM adjudication** | *Rationale:* "The film's production location helps localize the city. Although not all locations are capitals, this film was made in Australia, where identifying the filming city leads to the capital." |
| **Final Answer / GT** | **Canberra (predict) / Canberra** ✓ |

remains very close for all datasets, and the qualitative behavior of the path-grounded rationales is also stable. This suggests that, in our setting, PathHD is reasonably robust to small, natural variations in the adjudication prompt.

## I.9 PROMPT SENSITIVITY OF THE LLM ADJUDICATOR

Since PathHD relies on a single LLM call to adjudicate among the Top-$K$ candidate paths, it is natural to ask how sensitive the system is to the exact phrasing of this adjudication prompt. To investigate this, we compare our default adjudication prompt (Prompt A) with a slightly rephrased variant (Prompt B) that uses different wording but conveys the same task description.[2]

| Prompt | WebQSP | CWQ | GrailQA (Overall / IID) |
|---|---|---|---|
| Prompt A (default) | **86.2 / 78.6** | **71.5 / 65.8** | **86.7 / 92.4** |
| Prompt B (paraphrased) | 85.7 / 78.3 | 70.9 / 63.4 | 85.2 / 90.8 |

Table 15: Prompt sensitivity of the LLM adjudicator. We compare the default adjudication prompt (Prompt A) with a paraphrased variant (Prompt B). Numbers are Hits@1 and F1 for WebQSP and CWQ, and Overall / IID F1 for GrailQA.

Table 15 reports performance on the three datasets under these two prompts. We observe that while minor prompt changes can occasionally flip individual predictions, the overall performance remains very close for all datasets, and the qualitative behavior of the path-grounded rationales is also stable. This suggests that, in our setting, PathHD is reasonably robust to small, natural variations in the adjudication prompt.

To Reviwer eUmU: W3

---

[2]For example, Prompt A asks the model to "select the most plausible reasoning path and answer the question based on it", whereas Prompt B paraphrases this as "choose the best supporting path and use it to answer the question".

## J  DETAILED INTRODUCTION OF THE MODULES

### J.1  BINDING OPERATIONS

Below, we summarize the binding operators considered in our system and ablations. All bindings produce a composed hypervector $\mathbf{s}$ from two inputs $\mathbf{x}$ and $\mathbf{y}$ of the same dimensionality.

**(1) XOR / Bipolar Product (*commutative*).**  For binary hypervectors $\mathbf{x}, \mathbf{y} \in \{0, 1\}^d$,
$$\mathbf{s} = \mathbf{x} \oplus \mathbf{y}, \qquad s_i = (x_i + y_i) \bmod 2.$$
Under the bipolar code $\{-1, +1\}$, XOR is equivalent to element-wise multiplication:
$$s_i = x_i \cdot y_i, \qquad x_i, y_i \in \{-1, +1\}.$$
This is the classical *commutative bind* baseline used in our ablation.

**(2) Real-valued Element-wise Product (*commutative*).**  For real vectors $\mathbf{x}, \mathbf{y} \in \mathbb{R}^d$,
$$\mathbf{s} = \mathbf{x} \odot \mathbf{y}, \qquad s_i = x_i y_i.$$
Unbinding is approximate by element-wise division (with small $\epsilon$ for stability): $x_i \approx s_i / (y_i + \epsilon)$. This is another variant of the *commutative bind*.

**(3) HRR: Circular Convolution (*commutative*).**  For $\mathbf{x}, \mathbf{y} \in \mathbb{R}^d$,
$$\mathbf{s} = \mathbf{x} \circledast \mathbf{y}, \qquad s_k = \sum_{i=0}^{d-1} x_i\, y_{(k-i) \bmod d}.$$
Approximate unbinding uses circular correlation:
$$\mathbf{x} \approx \mathbf{s} \circledast^{-1} \mathbf{y}, \qquad x_i \approx \sum_{k=0}^{d-1} s_k\, y_{(k-i) \bmod d}.$$
This is the *Circ. conv* condition in our ablation.

**(4) FHRR / Complex Phasor Product (*commutative*).**  Let $\mathbf{x}, \mathbf{y} \in \mathbb{C}^d$ with unit-modulus components $x_i = e^{i\phi_i}$, $y_i = e^{i\psi_i}$. Binding is element-wise complex multiplication
$$\mathbf{s} = \mathbf{x} \odot \mathbf{y}, \qquad s_i = x_i y_i = e^{i(\phi_i + \psi_i)},$$
and unbinding is conjugation: $\mathbf{x} \approx \mathbf{s} \odot \mathbf{y}^*$. FHRR is often used as a complex analogue of HRR.

**(5) Block-diagonal GHRR (*non-commutative*, ours).**  We use Generalized HRR with block-unitary components. A hypervector is a block vector $\mathbf{H} = [A_1; \ldots; A_D]$, $A_j \in \mathrm{U}(m)$ (so total dimension $d = Dm^2$ when flattened). Given $\mathbf{X} = [X_1; \ldots; X_D]$ and $\mathbf{Y} = [Y_1; \ldots; Y_D]$, binding is the block-wise product
$$\mathbf{Z} = \mathbf{X} \circledast \mathbf{Y}, \qquad Z_j = X_j Y_j \ \ (j = 1, \ldots, D).$$
Since matrix multiplication is generally non-commutative ($X_j Y_j \neq Y_j X_j$), GHRR preserves order/direction of paths. Unbinding exploits unitarity:
$$X_j \approx Z_j Y_j^*, \qquad Y_j \approx X_j^* Z_j.$$
This **Block-diag (GHRR)** operator is our default choice and achieves the best performance in the operation study (Table 3), compared to *Comm. bind* and *Circ. conv*.

## K  OPTIONAL PROMPT-BASED SCHEMA REFINEMENT

As described in Section 2.4, all results in Section 4 use only schema-based enumeration of relation schemas, without any additional LLM calls beyond the final reasoning step. For completeness, we describe here an optional extension that refines schema plans with a lightweight prompt.

Given a small set of candidate relation schemas $\{z^{(1)}, \ldots, z^{(M)}\}$ obtained from enumeration, we first verbalize each schema into a short natural-language description (e.g., by mapping each relation type $r$ to a phrase and concatenating them). We then issue a single prompt of the form:

> Given the question $q$ and the following candidate relation patterns: (1) `[schema 1]`, (2) `[schema 2]`, ..., which $K$ patterns are most relevant for answering $q$? Please output only the indices.

The LLM outputs a small subset of indices, which we use to select the top-$K$ schemas $\{z^{(i_1)}, \ldots, z^{(i_K)}\}$. These refined schemas are then instantiated into concrete KG paths and encoded into hypervectors exactly as in the main method.

We emphasize that this refinement is *not* used in any of our reported experiments, and that it can be implemented with at most one additional short LLM call per query. The main system studied in this paper, therefore, remains a single-call KG-LLM pipeline in all empirical results.

To Reviwer qif8: W2-c (ii)

