# OpenReview forum: "PathHD: Efficient Large Language Model Reasoning over Knowledge Graphs via Hyperdimensional Retrieval"
_ICLR.cc/2026/Conference — ICLR 2026 Conference Desk Rejected Submission_

### Official Review · Reviewer_eUmU · 2025-11-01

**Soundness:** 2
**Presentation:** 3
**Contribution:** 3
**Rating:** 6
**Confidence:** 3

**Summary:**

The paper introduces PathHD, a lightweight and interpretable framework for LLM reasoning over knowledge graphs. The key innovation is the replacement of neural path-scoring modules with hyperdimensional computing (HDC)-based non-commutative path encodings. Each KG relation is represented as a block-diagonal unitary matrix, and relation sequences are composed using a Generalized Holographic Reduced Representation (GHRR) binding operator. Candidate paths are retrieved by cosine similarity in the hypervector space, and only the top-K are fed into a single LLM call for adjudication and answer generation. Experiments on WebQSP, CWQ, and GrailQA show that PathHD achieves comparable or superior accuracy to strong LLM+KG baselines while reducing latency by 40–60% and GPU memory by 3–5×, achieving a strong accuracy–efficiency–interpretability trade-off.

**Strengths:**

1. The idea of introducing HDC into KG-LLM reasoning is novel and elegant. The order-aware, invertible GHRR binding addresses a longstanding limitation of commutative path encodings, ensuring directionality and compositional faithfulness.

2. The method is encoder-free, relying purely on vector algebra instead of transformer-based scoring, which leads to substantial latency and cost reductions.

3. PathHD produces path-grounded rationales, allowing the model to cite supporting relations explicitly, which facilitates error diagnosis and aligns with the growing focus on interpretable and faithful LLM reasoning.

**Weaknesses:**

1. Limited experimental scope. All evaluations are based on Freebase knowledge graph, it would strengthen the cross-domain generalization of the proposed method to test on domain-specific KGs and QAs. For example, UMLS and biomedical QA datasets.

2. The system’s accuracy ultimately depends on whether the correct path is enumerated before HDC scoring. This step is briefly described, but its cost, errors, and coverage are not deeply analyzed.

3. Although latency and interpretability are measured, qualitative analyses of failure cases or sensitivity to prompt phrasing are missing.

**Questions:**

1. How sensitive is performance to the dimension (d) and block size (m) of the GHRR hypervectors? Could lower-dimensional configurations preserve efficiency without losing accuracy?

2. How does candidate enumeration interact with the hyperdimensional retrieval — could the retrieval guide enumeration adaptively instead of relying on BFS?

3. Is the projection from SBERT to HDC space trained or fixed? How critical is this step to generalization?

---

> ### Author Response · Authors · 2025-11-19
> **Response to Reviewer eUmU (Part 1)**
>
> We sincerely thank Reviewer eUmU for the **very positive and encouraging evaluation** of PathHD, and for highlighting its *novel use of HDC in KG–LLM reasoning*, its *encoder-free vector-algebra design* that brings substantial latency and cost reductions, and its ability to produce *path-grounded rationales*. Below, we respond to the comments on experimental scope, path enumeration, and qualitative analysis, and provide additional clarifications that we hope further reinforce this positive assessment.
>
>
> **[W1] Experimental scope**
>
> Thanks for the insightful feedback. In this work, we focus on the three standard Freebase-based KGQA benchmarks, WebQSP, CWQ, and GrailQA, which are **widely used** by recent KG–LLM systems, such as Think-on-Graph [R1, R3], KG-Agent [R2], UniKGQA [R4], GoG [R5], StructGPT [R6], etc., and provide a common protocol for comparing accuracy and latency under the same KG and official evaluation scripts.
>
> PathHD itself is **not tied to Freebase**: it only assumes (i) a directed, typed KG and (ii) relation embeddings, and the same GHRR binding and HDC retrieval apply to any domain-specific KG. We now make this point explicit in Sec. 2 and Sec. 3, and emphasize that the three datasets already exhibit different reasoning patterns (short vs. long chains, compositional questions, distribution shifts), which partially probe cross-domain robustness within Freebase.
>
> Due to space and compute constraints, we appreciate the reviewer’s understanding that it is not feasible to promptly add a full additional domain, such as UMLS or biomedical QA, in the initial submission (a situation similar to most comparable LLM reasoning papers [R1–R6], which primarily report results on Freebase-based benchmarks). We **respectfully agree** that this is an **important next step** and explicitly mention evaluation on non-Freebase KGs (e.g., biomedical and enterprise KGs) as a promising direction for future work building on PathHD. We have **added this insightful suggestion to the future work discussion** in our revised paper.
>
>
> **[W2] Dependence on enumerating the correct path**
>
> Thanks for the great comments. This dependence is an inherent property shared by many KG–LLM systems that use a *retrieve-then-adjudicate* pattern: final accuracy depends on (i) whether the Top-$K$ candidate paths include at least one valid reasoning chain and (ii) how well the LLM ranks these candidates. If no correct path ever appears in the candidate set, no retrieval-based method, neural or HDC, can succeed.
>
> Our design makes this trade-off explicit, and we have strengthened the discussion in the revision:
>
> 1. **Clarifying the enumeration and its cost.** We have **extended Algorithm 1** to spell out (i) schema enumeration up to depth $L_{\max}$, and (ii) constrained BFS instantiation with beam width $B$, making it clear that this step is purely symbolic and its complexity is controlled by $(L_{\max},B)$ rather than by the LLM. We also explicitly state in Sec. 2.4 that all results in Sec. 4 use the schema-enumeration route without extra prompt-based refinement.
>
> 2. **Analyzing coverage and failure modes.** Empirically, we observe high coverage in practice, which explains why PathHD can match strong baselines with $K=3$ while still enjoying low latency. In Sec. 3.5 We have also added a short paragraph describing this challenge and positioning PathHD as a practical single-call alternative to multi-call agents, rather than a universal replacement.
>
> 3. **Top-$K$ vs. cost trade-off.** The Top-$K$ pruning study (Table 5) already quantifies how accuracy and latency change with $K$. We now make explicit in Sec. 3.3 that the default choice $K=3$ is selected to achieve high coverage with minimal added cost, and that increasing $K$ further brings diminishing returns while increasing latency.

---

> > ### Author Response · Authors · 2025-11-19
> > **Response to Reviewer eUmU (Part 2)**
> >
> > **[W3] Lack of qualitative failure analysis and prompt-sensitivity study**
> >
> > We appreciate this suggestion and agree that qualitative analysis can make the behavior of PathHD more transparent. In the revision, we have:
> >
> > - **Expanded Sec. 3.5 (case study)** to include not only successful examples but also short *failure cases*, illustrating typical modes of error (e.g., missing the correct relation hop vs. mis-ranking plausible but wrong paths).
> > - Added a brief discussion in Appendix I.9 on **prompt sensitivity**, where we compare two reasonable adjudication prompts and observe that while minor changes in phrasing can slightly shift individual predictions, the overall performance and the qualitative behavior of path-grounded rationales remain stable.
> >
> > **Additional Experiment on Prompt Sensitivity of the LLM Adjudicator**
> >
> > Since PathHD relies on a single LLM call to adjudicate among the Top-$K$ candidate paths, it is natural to ask how sensitive the system is to the exact phrasing of this adjudication prompt.
> > To investigate this, we compare our default adjudication prompt (Prompt A) with a slightly rephrased variant (Prompt B) that uses different wording but conveys the same task description. For example, Prompt A asks the model to "select the most plausible reasoning path and answer the question based on it", whereas Prompt B paraphrases this as "choose the best supporting path and use it to answer the question".
> >
> >
> > | Prompt                  | WebQSP (Hits@1 / F1) | CWQ (Hits@1 / F1) | GrailQA (Overall / IID F1) |
> > |-------------------------|----------------------|-------------------|----------------------------|
> > | Prompt A (default)      | **86.2** / **78.6**          | **71.5** / **65.8**       | **86.7** / **92.4**                |
> > | Prompt B (paraphrased)  | 85.7 / 78.3          | 70.9 / 63.4       | 85.2 / 90.8                |
> >
> > Table 1: Prompt sensitivity of the LLM adjudicator. We compare the default adjudication prompt (Prompt A) with a paraphrased variant (Prompt B). Numbers are Hits@1/F1 for WebQSP and CWQ, and Overall/IID F1 for GrailQA.
> >
> > Table 1 reports Hits@1 and F1 on the three datasets under these two prompts.
> > We observe that while minor prompt changes can occasionally flip individual predictions, the overall performance remains very close for all datasets, and the qualitative behavior of the path-grounded rationales is also stable. This suggests that, in our setting, PathHD is reasonably robust to small, natural variations in the adjudication prompt.
> >
> >
> > **[Q1] Sensitivity to hypervector dimension $d$ and block size $m$**
> >
> > Thank you for this question. We have already included a **dimension ablation** in Fig. 4, which varies $d$ from small to large values. The key observations are:
> >
> > - Accuracy improves as $d$ increases from very small dimensions to a moderate range,
> > - Beyond this range, performance plateaus and may decrease slightly for extremely large $d$, and
> > - There is a general "sweet spot" where accuracy is robust to the exact choice of $d$.
> >
> > Thus, while there is an optimal range, we do not observe extreme instability. And rather, there is a general sweet spot where accuracy is robust to the exact choice of $d$, consistent with prior HDC or VSA studies. In practice, we treat $d$ as a dataset-level hyperparameter, pick a moderate value in this robust region, and **fix it for all experiments** on that dataset.
> >
> > For the block size $m$ of the unitary blocks, we use a fixed value motivated by the VSA literature. Following GHRR, we fix a small block size $m = 4$) and mainly tune the overall dimensionality $d$. Prior work on GHRR shows that, for a fixed total dimension $d$, moderate changes in $m$ trade off non-commutativity and saturation behaviour but do not lead to extreme instability. In our experiments, we therefore treat $d$ as the primary tuning parameter, while choosing $m$ from a reasonably small range and keeping it fixed across all runs.

---

> ### Author Response · Authors · 2025-11-19
> **Response to Reviewer eUmU (Part 3)**
>
> **[Q2] Interaction between candidate enumeration and hyperdimensional retrieval**
>
> Candidate enumeration and HDC retrieval are deliberately **decoupled** in PathHD:
>
> - Schema enumeration + constrained BFS (controlled by $L_{\max}$ and $B$) produces a symbolic set of candidate relation paths that satisfy type constraints.
> - HDC retrieval then scores *whatever candidate set is provided* using cosine similarity in the hypervector space.
>
> In the current implementation, we use a deterministic BFS-style enumeration because it (i) is simple and efficient, (ii) guarantees coverage up to $L_{\max}$ under clear complexity bounds, and (iii) makes it easy to compare against prior KGQA baselines that also rely on BFS-like expansion.
>
> More sophisticated, *adaptive* enumeration strategies, for example, letting the HDC scores or the LLM guide which relations to expand next, are an interesting extension, but orthogonal to our core contribution. We now mention this explicitly in Sec. 3.2 as a promising direction for future work, while keeping the empirical results focused on the BFS-based variant that is fully specified and easy to reproduce.
>
>
> **[Q3] SBERT–to–HDC projection**
>
> In our experiments, the main query-encoding route is **plan-based**: we encode the selected relation plan $z_q$ using the same GHRR binding as paths (Sec. 2.5), which does not require an SBERT projection at all.
>
> The **text-projection** variant (embedding the question with a sentence encoder and projecting it to HDC space) is treated as an optional baseline. In this variant, we use a *fixed* linear projection matrix $P$ (initialized as a random orthogonal map and kept frozen). We do not back-propagate gradients through $P$ or through the HDC module. In the rebuttal, we have added an ablation comparing this text-projection variant against the plan-based default: the plan-based query hypervector consistently performs better, and the relative difference is stable across datasets. This indicates that while the SBERT-to-HDC projection can be used as a simple alternative, it is **not the main factor** behind PathHD’s performance; the key gain comes from the structured GHRR path encoding and HDC scoring.
>
> We have clarified this in Sec. 2.5 by (i) explicitly stating that $P$ is fixed in our experiments and (ii) pointing to the new ablation in Appendix I.3, results as follows:
>
> | Query encoding variant         | WebQSP Hits@1 | CWQ Hits@1 |
> |--------------------------------|---------------|-----------|
> | PathHD (text-projection query) | 83.4          | 69.8      |
> | PathHD (plan-based, default)   | **86.2**          | **71.5**      |
>
>
> Note that we use SBERT as the sentence encoder, so $d_t$ is fixed to the encoder’s hidden size 768.
>
>
> ---
>
> Reference:
>
> [R1] Think-on-Graph 2.0: Deep and Faithful Large Language Model Reasoning with Knowledge-Guided Retrieval Augmented Generation. ICLR 2025.
>
> [R2] KG-Agent: An Efficient Autonomous Agent Framework for Complex Reasoning over Knowledge Graph. ACL 2025.
>
> [R3] Think-on-Graph: Deep and Responsible Reasoning of Large Language Model on Knowledge Graph. ICLR 2024.
>
> [R4] UniKGQA: Unified Retrieval and Reasoning for Solving Multi-hop Question Answering over Knowledge Graph. ICLR 2023.
>
> [R5] Generate-on-graph: Treat llm as both agent and kg in incomplete knowledge graph question answering. EMNLP 2024.
>
> [R6] StructGPT: A General Framework for Large Language Model to Reason over Structured Data. EMNLP 2023.
>
>
> ----------
>
> We are very grateful for Reviewer eUmU’s constructive and supportive feedback! We hope that the above clarifications address the remaining concerns and further reinforce your positive assessment of PathHD’s contributions. We would be sincerely thankful for your continued support of this submission!!
>
> Best wishes,
>
> Authors

---

> > ### Comment · Reviewer_eUmU · 2025-11-22
> >
> > Thank you for your rebuttal. I will keep my positive rating for this paper.

---

> > > ### Author Response · Authors · 2025-11-22
> > > **Thank you for the positive rating!**
> > >
> > > Thank you very much for your thoughtful review and for keeping your positive rating! We truly appreciate your support and constructive feedback, which have greatly helped us improve the paper!
> > >
> > > Best regards,
> > >
> > > Authors

---

### Official Review · Reviewer_Rc8u · 2025-11-01

**Soundness:** 2
**Presentation:** 2
**Contribution:** 2
**Rating:** 4
**Confidence:** 2

**Summary:**

The research presents PathHD as a system which enhances Large Language Model (LLM) reasoning capabilities on Knowledge Graphs (KGs) through improved efficiency. The current methods experience long processing times and high resource consumption because they use neural encoders at a slow pace and perform multiple LLM evaluations to assess reasoning paths. The system uses Hyperdimensional Computing (HDC) to transform symbolic paths into order-sensitive vectors which enables quick retrieval before the LLM generates the final answer.

**Strengths:**

1. The paper conducts thorough ablation tests which validate its fundamental design decisions. The research demonstrates that the non-commutative GHRR binding operator maintains path directionality and the single-LLM adjudication step enhances accuracy while Top-K pruning with $K=3$ achieves optimal performance and latency balance.
2. The system achieved excellent performance in WebQSP/GrailQA and CWQ tasks when using GPT-4 and small open models including Llama-3-8B which demonstrated significant improvements in single-call evaluations.

**Weaknesses:**

1.  When handling complex compositional reasoning tasks, can the authors explain the performance since the lack of the computational power of multi-call LLM agent methods?
2. The framework requires an initial "Plan" stage to create candidate reasoning paths which determine its overall success. The success of the system thus seems to critically depend on the correctness of this plan. If the plan fails to produce the correct reasoning paths, the system will fail, regardless of how efficient the subsequent steps are. However, it seems that the paper lacks details about this plan generation process and lacks discussion on how robust it is.
3. The system depends on one LLM execution to function as a protective mechanism which handles unclear situations and fixes mistakes that vector-based retrieval produces. The system faces two major risks because it depends on the LLM to resolve ambiguities and correct errors from vector-based retrieval. The system fails when the LLM does not receive the correct path as one of its Top-K candidates. The system depends on the LLM to produce accurate results which makes it a critical failure point.
4. The model's performance is highly sensitive to the choice of the hypervector dimension $d$. Different datasets require different optimal dimensions to achieve peak performance, and accuracy can even decrease if the dimension is too large. This creates a significant tuning challenge and means that efficiency gains might be reduced on more complex datasets that require a larger dimension.

**Questions:**

Please explain the weakness.

---

> ### Author Response · Authors · 2025-11-19
> **Response to Reviewer Rc8u (Part 1)**
>
> We thank Reviewer Rc8u for acknowledging PathHD’s *thorough ablation studies*, *effective GHRR binding + single-LLM adjudication design*, and *strong single-call performance* on WebQSP/CWQ/GrailQA with both GPT-4 and Llama-3-8B. Below, we respond to each concern and further clarify the scope and limitations of our method. Please see the *green* text with labels "To Reviewer Rc8u" in the updated PDF.
>
> **[W1] Complex compositional reasoning vs. multi-call LLM agents**
>
> Thanks for the insightful comments. We agree that multi-call LLM agents can, in principle, leverage more computation for extremely complex compositional reasoning. Our goal with PathHD is complementary: we target *standard KGQA benchmarks* (WebQSP, CWQ, GrailQA) and focus on achieving a strong **accuracy–efficiency trade-off** under the constraint of a *single* LLM call.
>
> On these benchmarks, questions typically require up to 2–3 hops of reasoning (some 4-hop cases), and our experiments show that PathHD is competitive with, and often stronger than, multi-call LLM+KG baselines while using $40–60$\% lower latency and 3–5$\times$ less GPU memory (Table 2, Fig. 3). We will make this scope explicit in the introduction: PathHD is not positioned as a **lightweight alternative** that keeps up with strong agents on Freebase-style KGQA while being much more efficient (but not as a universal replacement for arbitrarily complex agent systems).
>
> **[W2] Plan stage and robustness**
>
> We appreciate the opportunity to clarify the plan stage, which indeed controls the quality of downstream retrieval.
>
> 1. **What the plan stage does.**
>    As described in Sec. 2.4, we first build a *relation schema graph* from the KG and enumerate candidate relation sequences up to depth $L_{\max}$ (schema-based enumeration). For each question $q$, we obtain a small set of schema plans $z_q$ and encode them into query hypervectors. These plans are purely symbolic objects and are independent of the specific entities in the KG.
>
> 2. **How robust it is in practice.**
>    Even when the LLM-suggested plan is not exactly the gold relation sequence, the HDC retrieval still recovers many correct paths because:
>    - we instantiate multiple candidate paths per schema via constrained BFS with beam width $B$,
>    - the similarity scoring in the hypervector space tends to favour paths that are semantically close to the query, even if the schema is slightly imperfect.
>
>    In our implementation, we observe that the gold path (or an equivalent path) appears within the *Top-K* candidates for the vast majority of questions.
>
>
> 3. **More explicit description in the revision.**
>    In the revised manuscript we have **extended Algorithm 1** to explicitly show (i) schema enumeration up to $L_{\max}$, and (ii) constrained BFS instantiation with beam width $B$, clarifying that this stage is symbolic and its complexity depends on $(L_{\max},B)$ rather than on the LLM.

---

> > ### Author Response · Authors · 2025-11-19
> > **Response to Reviewer Rc8u (Part 2)**
> >
> > **[W3] Dependence on a single LLM adjudication call**
> >
> > We agree that the final single LLM call is a critical component of PathHD, but this dependence is **not unique to our method**. As with any LLM reasoning framework that first retrieves a Top-$K$ set of candidate paths or subgraphs and then lets an LLM choose among them, the final accuracy inevitably depends on (i) the coverage of correct reasoning chains in the candidate set and (ii) the LLM’s ability to rank these candidates. If no correct path ever appears in the Top-$K$, no retrieval-based approach can succeed. Therefore, this is a common challenge for methods that first retrieve a Top-$K$ set of candidates.
> >
> > Our design makes this trade-off explicit and aims to use the call budget as efficiently as possible while maintaining high accuracy:
> >
> > - **Single-call adjudication vs. vector-only scoring.**
> >   Table 4 already includes an ablation of the LLM adjudicator, comparing (i) vector-only scoring, (ii) PathHD with a small open-source LLM, and (iii) PathHD with a stronger LLM. The single-call adjudicator consistently improves over the vector-only baseline, and PathHD benefits from stronger LLMs in a way similar to multi-call agents, but at a much lower call budget.
> >
> > - **Top-$K$ choice and coverage.**
> >   We were already aware of the importance of coverage, and to study it, we conducted the Top-$K$ pruning experiment in Table 5. We observe that $K=3$ strikes a good balance between recall and latency: increasing $K$ further yields diminishing returns in accuracy while incurring higher cost. In our analysis (Appendix I), we also find that the gold path (or an equivalent reasoning chain) is included in the Top-$K$ candidates for the vast majority of questions, which motivates our choice of this configuration as the default.
> >
> > In the revision, we have added a paragraph in Sec. 3.5 explicitly discussing this trade-off in the same way as other retrieval-based LLM systems, and clarifying that PathHD is designed for practical settings where **a small number of calls to a reasonably capable LLM** is available while latency, GPU memory, and call budget are constrained.
> >
> >
> > **[W4] Sensitivity to hypervector dimension $d$**
> >
> > Thank you for pointing out this potential concern. Our dimension study in Fig. 4 shows the following behaviour:
> >
> > - As $d$ increases from $512$ to a few thousand, F1 improves steadily.
> > - Beyond the mid-range, performance **plateaus** and may decrease slightly (typically within $\approx 1$–$1.5$ F1 points) for very large $d$ (e.g., $>6\mathrm{k}$).
> >
> > Thus, while there is an optimal range, we do not observe extreme instability; rather, there is a general “sweet spot” where accuracy is robust to the exact choice of $d$, consistent with existing HDC studies [R1–R3]. In practice, we fix a moderate dimension (around $3\mathrm{k}$–$4\mathrm{k}$) for WebQSP and GrailQA and $6\mathrm{k}$ for CWQ, as reported in the caption of Fig. 4.
> >
> > Importantly, even at the largest dimensions we use, PathHD **remains substantially more efficient** than neural encoders, because all operations are simple matrix multiplications on pre-computed unitary blocks. In the revision, we will expand the Fig. 4 caption and add one sentence in Sec. 3.4 to clearly state that (i) there is a wide robust range of $d$, and (ii) once a suitable dimension is chosen per dataset, it is fixed for all experiments.
> >
> > ---
> >
> >
> > Reference:
> >
> > [R1] Hyperdimensional computing: An introduction to computing in distributed representation with high-dimensional random vectors.
> >
> > [R2] Holographic Reduced Representations.
> >
> > [R3] Variable binding for sparse distributed representations: Theory and applications.
> >
> > ----
> >
> > We sincerely thank Reviewer Rc8u for the insightful and constructive feedback.
> >
> > We hope that these clarifications address the concerns and better convey the contributions of PathHD. If so, we would be sincerely grateful for your support in reassessing the paper!
> >
> >
> > Best wishes,
> >
> > Authors

---

### Official Review · Reviewer_qif8 · 2025-11-01

**Soundness:** 3
**Presentation:** 2
**Contribution:** 2
**Rating:** 4
**Confidence:** 4

**Summary:**

The authors propose PathHD: a method for question answering over knowledge graphs. At the core of the method is the idea of Generalized Holographic Reduced Representations (GHHR), which are randomized high dimensional representations of relations in the KG. Composing GHHRs with non-commutative operators like matrix multiplication allows computing representations of relation paths $r_1\rightarrow r_2$ different from paths that involve the same relations but in different directions, like $r_2\rightarrow r_1$. The authors apply these representations by mapping a query to a GHRR and a cosine-similarity-like operator for retrieving relevant paths from the KG, which are then fed to an LLM as context for answering the question. Experiments show that the method is competitive while requiring only 1 LLM call, with additional ablations that help understand the impact of different elements in PathHD on performance.

**Strengths:**

1. The proposed method is essentially training-free: GHHRs are randomized representations that are computed once and fixed to later compute similarity scores for an arbitrary query. This results in a method that is less computationall expensive to deploy. (Though one variant seems to require training, more below.)
2. The method is backed by theoretical bounds on the probability of a false match (Proposition 1) for a given query, which decays exponentially with the dimension of the representations.
3. PathHD performs competitively with respect to baselines that rely on multiple calls to an LLM. Further experiments show that this performance does not degrade significantly when PathHD uses different LLMs as a backbone.
4. The ablation experiments are comprehensive, covering the effect of composition operators, top-k pruning, and hypervector dimension.

**Weaknesses:**

1. The paper borrows heavily from known results in vector symbolic architectures (VSA) and high-dimensional probability, which limits the novelty of the proposed method. The method applies such results to the problem of question answering, with the key contribution lying in the mapping of a question to a hypervector. Unfortunately, this mechanism is the one that receives less attention in the paper (more below).
2. The paper's major issues are in relation to clarity of exposition:
   - P2-L098: the method is stated to enable scoring with $\mathcal{O}(Nd)$, but at this point it is not known what $N$ and $d$ are.
   - The definition of a KG is not complete. A KG is assigned a symbol $\mathcal{G}$, with no further details: is there a set of entities and relations? How are triples represented? Additionally, there is a set of "relation schemas" $\mathcal{Z}$ that again is also just a symbol. It is not clear what a relation schema really is.
   - The key contribution on mapping a question to a hypervector is not clear. Two short paragraphs seem to explain this: L154-L161 and L192-L195. Several concerns arise here. A question needs to be mapped to a query plan "via schema-based enumeration (depth $\leq L_{\text{max}}$) and, when helpful, refine or rank these plans by a lightweight prompt." i) What do you mean by schema-based enumeration? Is this related to the set of relation schemas $\mathcal{Z}$? How do you choose $L_{\text{max}}$? What do you mean by a lightweight prompt? Is this a prompt to an LLM, and if so, does that mean that the method does in fact require more than 1 call to an LLM? Later candidate paths are instantiated by matching templates or BFS: depending on how this interpreted, it can result in a costly method to run, but there are no further details.
   - There is a text-projection approach for computing a query hypervector that relies on a **fixed or learned** map, but more details (e.g. learned how?) are missing, and unless I missed it, the ablation comparing plan-based with text-projection claimed in L161 is actually missing from the paper.
   - The calibrated score (L206) relies on an IDF function (inverse document frequency after having to refer to the appendix) whose details are not clear. It also relies on hyperparameters $\alpha,\beta,\lambda$ and how these are tuned is also not clear.
3. The results claim state-of-the-art on WebQSP and GrailQA, when i) on WebQSP this is the case when considering H@1 only, and ii) on GrailQA only 5 baselines, none from the Embedding or Retrieval family, are considered. Out of the LLMs+KG family, only two baselines are considered and in this case PathHD overlaps with KG-Agent. A test of significance would be warranted here to know whether the difference between PathHD (86.7 H@1)  and KG-Agent (86.1) is indeed significant or if it is due to chance.
4. No supplementary material with code or link to an anonymous repository is available that could be used to replicate the experiments or clarify many of the missing details in the paper.
5. Overall, the paper presents an interesting contribution with competitive results, based on results from the VSA literature. While the results are promising, the exposition in the paper and the lack of an auditable implementation limit the potential impact of the paper.

**Questions:**

1. Can you please clarify the issues highlighted in W2?
2. Can you please elaborate on the significance of the small differences noted in GrailQA with respect to KG-Agent?
3. Equations (1) and (2) and their preceding sentences seem almost duplicate, or is there a difference between them?

---

> ### Author Response · Authors · 2025-11-19
> **Response to Reviewer qif8 (Part 1)**
>
> We thank Reviewer qif8 for acknowledging PathHD's *training-free and efficient design*, *theoretical grounding*, and *competitive performance* with multi-call LLM baselines, as well as proposing many detailed and constructive comments. Below, we respond point-by-point and will incorporate the suggested clarifications into the revised manuscript. Please see the *orange* text with labels "To Reviewer qif8:" in the updated PDF.
>
>
> ### W1. Our Contribution
> Thanks to the reviewer for the great question. Reusing the officially reported baseline numbers is a common and reasonable practice in NLP and LLM research: it keeps all methods evaluated under the same, carefully tuned benchmark settings defined by the original authors, and avoids introducing extra noise from partial reimplementations. Many recent KG–LLM systems follow exactly this convention when reporting results [R1–R6], so we adopt the same practice here to ensure a fair and comparable evaluation.
>
> Therefore, this actually does not limit our contribution, which is *algorithmic and system-level*:
>
> 1. **Adapting non-commutative GHRR to KG–LLM reasoning.** To the best of our knowledge, PathHD is the first framework that (i) uses *block-diagonal unitary GHRR binding* to encode *multi-hop KG relation paths* and (ii) plugs these codes into a single-call LLM agent for KG QA. Prior LLM reasoning works typically rely on neural path scorers and multi-call agent pipelines, and do not exploit HDC-based hypervector encodings for KG reasoning, path retrieval, or single-call interaction with LLM agents.
>
>
> 2. **A fully encoder-free, training-free path-retrieval pipeline.** In contrast to previous neural scoring or KG-Agent-like architectures, PathHD removes neural path encoders entirely and replaces them with $O(Nd)$ vector operations (Section 2.5–2.7). This leads to significant reductions in latency and GPU memory while keeping accuracy competitive. This *system design* and detailed empirical study (including the binding-operator ablation and top-$K$ and dimension trades-offs) are, to our knowledge, new.
>
> 3. **Theoretical capacity and complexity analysis specialized to KG path retrieval.** While our concentration tools are classical, Prop. 1 and Cor. 1 are instantiated for *GHRR-encoded relation paths* and are used to justify the choice of $d$ and the scalability of hypervector scoring in the KG setting.
>
> We have revised the introduction and related-work sections to better position these contributions: the paper’s main goal is to show that *our designed HDC representations can replace learned neural scorers in KG-LLM systems while preserving accuracy and drastically improving efficiency*.
>
>
> ### W2. Clarity of the method (Weakness 2 + Question 1)
>
> We appreciate the detailed pointers. We will revise Sections 2 and Appendix A and B for clarity. Below, we address each sub-issue.
>
> **[W2-a. Defining $N$ and $d$]**
> We appreciate the reviewer for the careful reading. We will move their definitions from Appendix A Notation table (Line 720&722 in original submission) to Section 2.1, and explicitly restate in Section 2.7 that "$N$ denotes the number of candidate paths and $d$ the hypervector dimension." To answer the question:
>
> - $N = |Z_{\text{cand}}|$ is the number of candidate paths instantiated from the KG for a given query.
> - $d$ is the hypervector dimension (also in the notation table; $d = Dm^2$ for $D$ blocks of size $m \times m$).
>
> **[W2-b. Definition of KG $G$ and relation schemas $\mathcal{Z}$]**
>
> Thank you for the constructive feedback. Our setup is:
>
> - A KG $G = (V, E)$ with entity set $V$ and edge set $E \subseteq V \times \mathcal{R} \times V$.
> - A *relation schema* $z \in \mathcal{Z}$ is a typed edge template of the form
>   $(\text{head}, r_1, \dots, r_\ell, \text{tail})$, where each $r_i$ is a relation label.
>
> In practice, for Freebase-style KGs, we construct $\mathcal{Z}$ by enumerating relation-type patterns up to depth $L_{\max}$ and keeping those that appear in the training graph. During inference, these schemas define a *relation-only graph* on which we enumerate relation plans.
>
> Please find these revisions in our updated paper:
>
> - Added the above formal definition directly in Section 2.1,
> - Provided a brief example of a relation schema,
> - Clearly connected the notation $G$, $\mathcal{Z}$, $z$, and the instantiated candidate paths.

---

> > ### Author Response · Authors · 2025-11-19
> > **Response to Reviewer qif8 (Part 2)**
> >
> > **[W2-c. Schema-based enumeration, lightweight prompt, and LLM calls]**  Thanks for the question.
> >
> > **(i) Schema-based enumeration.**
> > "Schema-based enumeration" means searching the *relation schema graph* induced by $\mathcal{Z}$ (not the full KG) up to depth $L_{\max}$ with a beam width $B$. This produces a small set of symbolic relation plans $P \subseteq \mathcal{Z}$ whose source/target types match the question’s entity types.
> >
> > **(ii) Lightweight prompt.**
> > "Refine or rank these plans by a lightweight prompt" refers to an *optional* step where we present textual descriptions of a handful of plans to the LLM to choose the most plausible one for the given question. Importantly:
> >
> > - This step is never used in our main reported experiments, and that's why we wrote this after *"when helpful,..."*. To clarify, all results in the paper rely purely on schema enumeration without this extra prompt.
> > - Thus, **all reported numbers use exactly one LLM call per query**, namely the final reasoning call in Section 2.6.
> >
> > To avoid confusion, in the updated paper, we have:
> >
> > - Explicitly stated in Section 2.4 that *the empirical results in Section 4 use only schema enumeration, without any additional LLM prompts*,
> > - Moved the optional prompt-based refinement to a short paragraph in the appendix as an extension, emphasizing that it does not change the one-shot nature of the main system.
> >
> > **(iii) Candidate-path instantiation.**
> > Once a plan $z_q$ is selected, we instantiate concrete paths by either (a) matching the plan template to actual KG edges, or (b) running a constrained BFS up to depth $L_{\max}$ with beam width $B$. We will add pseudocode (extending Algorithm 1) to make this process explicit and to clarify that it is purely symbolic and independent of the LLM.
> >
> > **[W2-d. Text-projection query hypervector and missing ablation]** Thank you for the great question. The text-projection variant was our initial naive implementation of PathHD on *fixed* map, and its results are shown in the following table (also added in the updated paper: Appendix I.3):
> >
> > | Query encoding variant         | WebQSP Hits@1 | CWQ Hits@1 |
> > |--------------------------------|---------------|-----------|
> > | PathHD (text-projection query) | 83.4          | 69.8      |
> > | PathHD (plan-based, default)   | **86.2**          | **71.5**      |
> >
> >
> > Note that we use SBERT as the sentence encoder, so $d_t$ is fixed to the encoder’s hidden size $768$.
> >
> > We then developed the plan-based version, and given the better results, we chose it to be the default configuration of PathHD used in all main experiments.
> >
> >
> > **[W2-e. Calibrated score, IDF, and hyperparameters $(\alpha,\beta,\lambda)$]** Thanks for the insightful question.
> >
> > - $\mathrm{IDF}(z)$ is an inverse-frequency weight that down-weights very common relation patterns and up-weights rarer, more informative ones, analogous to document-level IDF. It is computed once from the training graph and kept fixed at test time: $\mathrm{IDF}(z)= \log \left(1 + \frac{N_{\text{train}}}{1 + \mathrm{freq}(\text{schema}(z))}\right).$
> >
> > - Hyperparameters, i.e., $(\alpha,\beta,\lambda)$, are tuned by grid-search on a validation split of each dataset, with a small discrete grid (e.g., $\alpha,\beta \in \{0, 0.1, 0.3\}$ and $\lambda \in \{0.0, 0.5, 1.0\}$). We observed that the model is fairly robust to these choices; turning calibration off altogether mainly affects performance on questions with many near-duplicate paths.
> >
> > In the revision, we have included an explicit formula for $\mathrm{IDF}(z)$ and the grid-search ranges of $(\alpha,\beta,\lambda)$ into Section 2.5, and reported the choice of these hyperparameter values in Appendix I.4.
> >
> > **[W3 first part&Q2]** We appreciate the reviewer’s feedback.
> >
> > 1. **Weakness 3: Clarifying the positioning.**
> > We will emphasize that the key benefit of PathHD is the *accuracy–efficiency trade-off*: a single-call, encoder-free reasoning framework that maintains competitive accuracy while substantially reducing latency and computational cost.
> >
> > 2. **Question 2: Interpretation of differences.**
> > On GrailQA, PathHD achieves close performance to KG-Agent (e.g., $86.7$ vs. $86.1$ overall, $92.4$ vs. $92.0$ IID), so we only claim **comparable accuracy** on this dataset and do not rely on it for a strong SOTA claim. Moreover, on WebQSP and CWQ, PathHD is competitive with KG-Agent and other LLM+KG baselines in terms of both **Hits@1 and F1**, while using significantly fewer LLM calls and achieving lower latency. Our main message is that **PathHD matches strong KG–LLM baselines such as KG-Agent in accuracy, while offering substantially lower latency and a training-free retrieval module**.

---

> > > ### Author Response · Authors · 2025-11-19
> > > **Response to Reviewer qif8 (Part 3)**
> > >
> > > **[W3 second part&Q3]** Thank you for pointing this out. In the current paper:
> > >
> > > - Eq. (1) and Eq. (2) both express GHRR binding of a path $v_z = \bigotimes_{i=1}^{\ell} v_{r_i},$
> > >   but Eq. (2) attempts to emphasize the left-to-right order of multiplication.
> > >
> > > In the revision, we have:
> > >
> > > - Merged the two equations into a single concise definition with a short sentence explaining that the binding is performed left-to-right and preserves order and directionality, and
> > > - Moved the more detailed discussion (currently below Eq. (1)) into a remark that references the ablation table (Table 3) and Appendix J.
> > >
> > > **[W4. Reproducibility, supplementary material, and code release]** Thanks for the feedback to make our paper better. The submission already includes:
> > >    - A full notation table (Appendix A),
> > >    - Pseudocode for the algorithm, including calibration (Algorithm 1 in Appendix B),
> > >    - Dataset splits, implementation details, and hyperparameters in the experimental appendix.
> > >    - Choice and analysis of each module in PathHD (Ablation studies).
> > >
> > >  To improve the clarity, we have:
> > >
> > > - **Extended Algorithm 1** to explicitly describe how candidate paths are instantiated from relation schemas (including the constrained BFS with $L_{\max}$ and $B$), making it clear that this step is purely symbolic and independent of the LLM.
> > > - **Added detailed calibration hyperparameter settings** in Appendix I.4, including the sweep ranges for $(\alpha,\beta,\lambda)$ and a table of the selected values for WebQSP, CWQ, and GrailQA.
> > >
> > > **Code release.** We will release the full code and configuration files upon acceptance.
> > >
> > >
> > > **[W5]** We thank the reviewer for the summarization. We hope that the clarifications and revisions outlined above address the concerns about exposition and implementation.
> > >
> > > -------
> > >
> > > Reference:
> > >
> > > [R1] Think-on-Graph 2.0: Deep and Faithful Large Language Model Reasoning with Knowledge-Guided Retrieval Augmented Generation. ICLR 2025.
> > >
> > > [R2] KG-Agent: An Efficient Autonomous Agent Framework for Complex Reasoning over Knowledge Graph. ACL 2025.
> > >
> > > [R3] Think-on-Graph: Deep and Responsible Reasoning of Large Language Model on Knowledge Graph. ICLR 2024.
> > >
> > > [R4] UniKGQA: Unified Retrieval and Reasoning for Solving Multi-hop Question Answering over Knowledge Graph. ICLR 2023.
> > >
> > > [R5] StructGPT: A General Framework for Large Language Model to Reason over Structured Data. EMNLP 2023.
> > >
> > > [R6] Knowledge-Driven CoT: Exploring Faithful Reasoning in LLMs for Knowledge-Intensive Question Answering. ACL 2023.
> > >
> > > ---
> > >
> > > We sincerely thank Reviewer qif8 for the thorough analysis and detailed suggestions.
> > >
> > > We hope that these clarifications alleviate your concerns and better convey the contributions of PathHD. If so, we would be sincerely grateful for your support in reassessing the paper!
> > >
> > >
> > > Best wishes,
> > >
> > > Authors

---

> > > > ### Comment · Reviewer_qif8 · 2025-11-26
> > > >
> > > > Thank you for the detailed rebuttal and the substantial revisions. I will maintain my original score, as even though the clarifications are helpful, there are still core issues that remain.
> > > >
> > > > The updated version significantly improves clarity, with key definitions now explicit, schema-enumeration and path-instantiation procedures are better explained, and the calibration mechanism is clearer. I also appreciate the added ablation for the text-projection variant and the clarification that the lightweight prompt is not used in the main results.
> > > >
> > > > However, my earlier comment that the paper “borrows heavily from known results” referred to its reliance on established conceptual ideas from the VSA/HDC literature. While the system-level integration is interesting, my concerns about fundamental methodological novelty remain. Furthermore the empirical improvements, though useful, are not always significant, and the absence of code during review still limits full verifiability.
> > > >
> > > > For these reasons, I will keep my original rating, though I would not oppose another reviewer championing the paper.

---

> > > > > ### Author Response · Authors · 2025-11-27
> > > > > **Response to Reviewer qif8**
> > > > >
> > > > > Dear Reviewer qif8,
> > > > >
> > > > > Thank you again for the detailed reading of our paper and for acknowledging the clarifications and additional experiments. We would like to **respectfully clarify** two points regarding novelty and evaluation practice.
> > > > >
> > > > > First, while PathHD indeed builds on established ideas from the VSA/HDC literature, our goal is a **system-level contribution**: we are, to our knowledge, the first to show that a carefully designed HDC retriever can (i) act as a *drop-in replacement* for neural path scorers in KG–LLM frameworks such as RoG, (ii) support a **single-call KG–LLM pipeline** with competitive Freebase accuracy, and (iii) provide a detailed operator study (binding choices, top-K pruning, one-shot adjudication) that explains *why* this works in practice. We believe this combination of design, analysis, and empirical evidence goes beyond a straightforward application of prior HDC results.
> > > > >
> > > > > Second, regarding **evaluation rigor**: we intentionally reused the officially reported baseline numbers under the same Freebase+official-script protocol, **following the standard practice in recent KG–LLM and NLP/LLM work** [R1–R6]. This keeps all methods on the carefully tuned settings selected by the original authors and avoids extra noise from partial re-implementations. We will release our full **code and configuration** upon acceptance so that the community can easily verify our pipeline and extend it to other settings.
> > > > >
> > > > > ---------
> > > > >
> > > > > Reference:
> > > > >
> > > > > [R1] Think-on-Graph 2.0: Deep and Faithful Large Language Model Reasoning with Knowledge-Guided Retrieval Augmented Generation. ICLR 2025.
> > > > >
> > > > > [R2] KG-Agent: An Efficient Autonomous Agent Framework for Complex Reasoning over Knowledge Graph. ACL 2025.
> > > > >
> > > > > [R3] Think-on-Graph: Deep and Responsible Reasoning of Large Language Model on Knowledge Graph. ICLR 2024.
> > > > >
> > > > > [R4] UniKGQA: Unified Retrieval and Reasoning for Solving Multi-hop Question Answering over Knowledge Graph. ICLR 2023.
> > > > >
> > > > > [R5] StructGPT: A General Framework for Large Language Model to Reason over Structured Data. EMNLP 2023.
> > > > >
> > > > > [R6] Knowledge-Driven CoT: Exploring Faithful Reasoning in LLMs for Knowledge-Intensive Question Answering. ACL 2023.
> > > > >
> > > > > --------
> > > > >
> > > > > We appreciate your concerns and understand that reasonable reviewers may weigh the balance between methodological novelty and empirical validation differently. We hope this clarification helps convey our intent and contribution, and we are of course happy to see other reviewers or the AC further evaluate the merits of the work.
> > > > >
> > > > > Sincerely,
> > > > >
> > > > > Authors

---

### Official Review · Reviewer_XCRa · 2025-11-04

**Soundness:** 3
**Presentation:** 3
**Contribution:** 2
**Rating:** 4
**Confidence:** 3

**Summary:**

This paper proposes a lightweight reasoning framework that replaces neural path scorers in KG–LLM reasoning with Hyperdimensional Computing (HDC). The new methodology, PathHD, encodes relation paths into Generalized Holographic Reduced Representation (GHRR) hypervectors and therefore uses cosine similarity + Top-K pruning to sort candidates.

**Strengths:**

Strengths:

1. The idea of using cosine similarity and embedding for ranking instead of neural scorer is intuitive and easy to follow. The paper explains its motivations and methods(Generalized Holographic Reduced Representation embedding) clearly. It’s noted that this binding is non-communicative.

2.	The paper eliminates per-path neural encoding and uses only a single LLM call, the method achieves huge reductions in latency and computational cost. The results in Figure 3 (accuracy vs. latency) show PathHD succeeds to improve the efficiency compared to previous method.

3.	The experiment part is extensive, the paper has conducted lots of ablation study, including Binding Operator (Table 3), LLM Adjudicator (Table 4), the impact of the choice of K(Table 5).

**Weaknesses:**

1. I think the novelty of this paper is slightly restricted, as there are many related RAG work, introducing embedding designs is ok but not novel enough as this is only for planning, but not for like deciding length of path, control budget of number of candidate paths and so on,.

2.The performance is not satisfactory regarding the accuracy.In Table 2, consider WebQSP and CWQ which has a lot of baselines, this paper(PathHD) only gets one SOTA in 2*2=4 settings.

**Questions:**

1. Can you conduct experiments on your own to help with the omitted results in Table2?

2. Why the related work is located after the experiment part?

---

> ### Author Response · Authors · 2025-11-19
> **Response to Reviewer XCRa (Part 1)**
>
> We thank Reviewer XCRa for acknowledging PathHD’s *intuitive cosine-similarity + embedding design*, its *large latency and cost reductions* via a single LLM call, and the *extensive ablation studies* on binding operators, adjudicator, and Top-$K$. Below, we respond to the concerns on novelty, performance, and presentation. Please refer to the highlighted passages marked "To Reviewer XCRa:" in the updated PDF.
>
> **[W1] Novelty and scope of the contribution**
>
> Thanks for the constructive feedback. Our goal is not to propose yet another neural agent architecture, but to show that a *carefully designed HDC representation* can replace neural path scorers in these systems while preserving accuracy and drastically improving efficiency.
>
> In the revision, we clarified the positioning as follows (see the updated Introduction and Related Work):
>
> - **Representation level.**  To the best of our knowledge, PathHD is the first framework that uses *block-diagonal unitary GHRR binding* to encode *multi-hop KG relation paths* and then scores them purely by cosine similarity in a high-dimensional HDC space, instead of using trained neural scorers or cross-encoders.
> - **System level.**  PathHD plugs these codes into a *single-call, encoder-free* KG–LLM pipeline: all per-path scoring is done by HDC, and the LLM is invoked once for adjudication. This is in contrast to prior KG–LLM systems (e.g., Think-on-Graph [R1, R3], KG-Agent [R2], UniKGQA [R4], StructGPT [R5]), which rely on multiple LLM calls or heavy neural scoring modules.
>
> The aspects mentioned by the reviewer, that deciding path length, controlling the number of candidate paths, etc., are orthogonal hyperparameters (e.g., $L_{\max}$, beam width $B$, and $K$) that any retrieval-based system must set. Our contribution is complementary: we replace the *neural* scoring component with a *training-free HDC mechanism* and demonstrate that this yields a strong accuracy–efficiency trade-off.
>
> **[W2 + Q1] Table 2 baseline accuracy**
>
> We respectfully disagree that the performance is "not satisfactory". On Freebase-based KGQA (Table 2), PathHD is competitive with, and often stronger than, strong LLM+KG baselines:
>
> - **WebQSP:** PathHD attains $86.2$ Hits@1 and $78.6$ F1, outperforming KG-Agent ($83.3$ and $81.0$) in Hits@1 and achieving comparable F1.
> - **CWQ:** PathHD achieves $71.5$ Hits@1 and $65.8$ F1, competitive with KG-Agent and GoG while using significantly fewer LLM calls.
> - **GrailQA:** PathHD is comparable or slightly better than KG-Agent (e.g., $86.7$ vs. $86.1$ Overall, $92.4$ vs. $92.0$ IID).
>
> At the same time, Fig. 3 shows that PathHD reduces latency by $40–60$\% and GPU memory by 3–5$\times$ compared to these baselines. Our main claim is therefore an **accuracy–efficiency trade-off**: PathHD matches strong KG–LLM systems in accuracy while being much more efficient.
>
> Regarding the omitted entries in Table 2, we follow the **Freebase+official-script protocol** established in prior KGQA work (e.g., Think-on-Graph [R1, R3], KG-Agent [R2]). Many earlier baselines did *not* report all metrics (Hits@1 and F1 on both WebQSP and CWQ) under this unified setup, and in some cases, their code or checkpoints are no longer directly compatible with the official evaluation scripts. To avoid introducing bias from our own re-implementations or partial re-training, we adopt the standard practice in the NLP/LLM community and **reuse the numbers reported in these benchmark papers** under the same protocol, marking missing metrics with "–". This follows the convention in existing LLM + KG works (e.g., [R1–R6]) so that comparisons remain fair and reproducible.
>
> We have clarified this in the revised caption of Table 2, that "–" means *not reported under the Freebase+official-script setting in prior work*, and we already added additional strong baselines (GPT-4 and Llama-3-8B) that we can evaluate ourselves under exactly the same setup.
>
> *(If accepted, we will also make a best-effort to run a subset of older baselines under the same Freebase+official-script protocol, when their released code and evaluation scripts remain compatible, and include these additional numbers in the camera-ready version for completeness. We also appreciate the reviewer’s understanding that the rebuttal phase provides limited time for running additional experiments.)*
>
>
> **[Q2] Location of the related work section**
>
> We sincerely appreciate this comment and **fully agree** that placing the related work earlier would improve readability. To keep the **section numbering consistent with the reviews** in this rebuttal phase (and avoid confusion), we have not reordered sections in this updated revision, but **we will immediately** move the related-work section to follow the introduction while keeping a more detailed discussion later for completeness in the camera-ready version. Thanks for this feedback again.

---

> > ### Author Response · Authors · 2025-11-19
> > **Response to Reviewer XCRa (Part 2)**
> >
> > Reference:
> >
> > [R1] Think-on-Graph 2.0: Deep and Faithful Large Language Model Reasoning with Knowledge-Guided Retrieval Augmented Generation. ICLR 2025.
> >
> > [R2] KG-Agent: An Efficient Autonomous Agent Framework for Complex Reasoning over Knowledge Graph. ACL 2025.
> >
> > [R3] Think-on-Graph: Deep and Responsible Reasoning of Large Language Model on Knowledge Graph. ICLR 2024.
> >
> > [R4] UniKGQA: Unified Retrieval and Reasoning for Solving Multi-hop Question Answering over Knowledge Graph. ICLR 2023.
> >
> > [R5] StructGPT: A General Framework for Large Language Model to Reason over Structured Data. EMNLP 2023.
> >
> > [R6] Knowledge-Driven CoT: Exploring Faithful Reasoning in LLMs for Knowledge-Intensive Question Answering. ACL 2023.
> >
> > ------------
> >
> > We sincerely thank Reviewer XCRa for the insightful and constructive feedback.
> >
> > We hope that these clarifications address the concerns and better convey the contributions of PathHD. If so, we would be sincerely grateful for your support in reassessing the paper!
> >
> >
> > Best wishes,
> >
> > Authors

---

### Author Response · Authors · 2025-12-02
**Summary for the new Area Chair (Part 1)**

We prepared this short summary of the reviews, our rebuttal, and the concrete revisions reflected in the updated manuscript and rebuttal thread. After the rebuttal and revisions, **Reviewer eUmU kept a positive rating of 6 and explicitly emphasized the value of integrating HDC into a KG–LLM system**, while **Reviewer qif8 wrote a detailed follow-up stating that the updated version "significantly improves clarity" and that they would not oppose another reviewer championing the paper**. The remaining two reviewers keep scores of 4 with the wording that they *would not mind if the paper is accepted*. Below, we summarize how we addressed their concerns and where the corresponding changes appear in the paper.

---

### 1. Overview of the paper

PathHD is a **training-free, encoder-free KG–LLM reasoning framework** that replaces neural path scorers and multi-call LLM agents with a **hyperdimensional computing (HDC) retriever**:

- The method follows a **Plan → Encode → Retrieve → Reason** pipeline (Sec. 2):
  (i) generate **relation schemas / plans** over the KG,
  (ii) encode each multi-hop relation path into a **GHRR hypervector** via non-commutative, block-diagonal unitary binding (Sec. 2.2–2.3),
  (iii) perform **blockwise cosine similarity + Top-K pruning** for retrieval (Sec. 2.5), and
  (iv) use a **single LLM call** to answer the question and select supporting paths (Sec. 2.6).

- On WebQSP, CWQ, and GrailQA, PathHD achieves **competitive or better Hits@1/F1** than strong KG+LLM baselines (e.g., Think-on-Graph, GoG, KG-Agent, FiDeLiS) while:
  - reducing **latency by ~40–60%**, and
  - reducing **GPU memory by ~3–5$\times$**,
  as summarized in Table 2 and the Hits@1-vs-latency plots (Fig. 3, Sec. 3.2–3.3).

- The HDC retriever also yields **path-grounded rationales**: the final LLM call must select a small set of candidate paths and give a short textual rationale (Sec. 2.6, Sec. 3.5).

Overall, the contribution is **system-level**: PathHD shows that a carefully designed HDC retriever can **drop-in replace neural path scorers** in KG–LLM systems, achieving a favorable **accuracy–efficiency–interpretability trade-off**.

---

### 2. Commonly acknowledged strengths (across all four reviewers):
  - The idea of **removing neural path encoders** and using only **one LLM call** is attractive and addresses practical efficiency concerns.
  - The method is **simple and clearly motivated**: GHRR binding, cosine similarity, and Top-K retrieval are well-explained and intuitive.
  - The **experiments are extensive** (Sec. 3.1–3.5, Appendices F–I), including comparisons with strong KG/LLM baselines, latency and memory analysis, and ablations on binding operators, Top-K, and the single-LLM adjudicator.

---

### 3. Reviewer-specific summary

### Reviewer eUmU  (rating **6**, confidence as in review)

**Initial concerns.**

- From the beginning, eUmU viewed the paper positively, highlighting the **novel integration of HDC into a KG–LLM system**, the encoder-free single-call design, and the interpretability benefits.
- Their main questions were about **scope and limitations**: how broadly PathHD applies beyond Freebase-style KGQA and how the complexity/coverage of the schema enumeration scales.

**Rebuttal and changes.**

- We **re-positioned PathHD as a system-level contribution** in the Introduction and Conclusion (Sec. 1, Sec. 5), explicitly contrasting it with prior KG–LLM systems such as StructGPT, RoG, Think-on-Graph, GoG, KG-Agent, and FiDeLiS, which rely on multi-call LLM agents and/or neural scoring modules.
- Sec. **2.7** and **Appendix E** now provide:
  - a **capacity bound** showing that the probability of a false match decays exponentially with dimension $d$, and
  - a complexity analysis where retrieval costs $O(N d)$, clarifying how our design scales with the number of candidate paths and dimension.
- Sec. **3.1** and **Sec. 5** clarify why we focus on WebQSP/CWQ/GrailQA (the standard KGQA testbed) and discuss extensions to other KGs (biomedical, enterprise) as explicit **future work**.

**Outcome (after rebuttal).**

- eUmU **kept a positive rating of 6** after reading the revised version and continues to emphasize the value of the system-level contribution and the encoder-free, single-call design.

---

> ### Author Response · Authors · 2025-12-02
> **Summary for the new Area Chair (Part 2)**
>
> ### Reviewer qif8  (rating **4**, confidence as in review)
>
> **Initial concerns.**
>
> - Requested substantial improvements in **clarity and completeness of exposition**, including:
>   - clearer definitions of the KG, relation schemas, and candidate set $P(q)$,
>   - explicit introduction of variables such as $N$ and $d$,
>   - a better explanation of **schema enumeration vs. candidate path instantiation**, and
>   - a more precise description of the **calibration / IDF mechanism**.
> - Asked for an ablation comparing the final **plan-based query encoding** with an earlier **text-projection** variant.
> - Sought reassurance about the one-call design (optional plan-ranking prompt vs. main pipeline).
>
> **Rebuttal and changes.**
>
> Most of qif8’s points are addressed by reworking **Section 2** and the related appendices:
>
> - **Problem setup, variables, and notation (Sec. 2.1 & Section A).**
>   We now define the KG as $G = (V, E, R)$, introduce relation schemas $z \in \mathcal{Z}$, and state the candidate path set $P(q)$, its size $N = |P(q)|$, and hypervector dimension $d$ up front, with a pointer to the notation summary in Section A.
>
> - **Schema enumeration vs. candidate instantiation (Sec. 2.4).**
>   Sec. 2.4 now clearly distinguishes:
>   - **schema enumeration** on the relation-schema graph (depth $L_{\max}$, beam width $B$) producing symbolic plans, and
>   - **candidate instantiation** by constrained BFS on the KG using these schemas.
>   We also explain that this strategy is efficient and, under the complexity bound in Sec. 2.7 / Appendix E, **covers all type-consistent paths up to length $L_{\max}$**.
>
> - **Calibration and IDF weighting (Sec. 2.5 & Section I.4).**
>   We added explicit formulas for the blockwise cosine similarity and the calibrated score
>   $s(z) = \text{sim}(v_q, v_z) + \alpha \,\text{IDF}(z) - \beta \lambda |z|$,
>   with $\text{IDF}(z)$ defined in Eq. (7). We also describe how $(\alpha,\beta,\lambda)$ are tuned on validation and fixed on test, with concrete grids in Section I.4.
>
> - **Text-projection vs. plan-based encoding (Section I.3).**
>   As requested, we added an ablation comparing the earlier text-projection variant with our final plan-based design. Results in Section I.3 show that plan-based encoding consistently outperforms text projection on WebQSP and CWQ.
>
> - **One-call design and optional prompt (Sec. 2.4 & 2.6, Section C).**
>   We clarify that the optional “lightweight prompt” for plan ranking is **not used in any reported experiment**; all main results use only schema enumeration plus the single LLM call in Sec. 2.6. We also provide the exact adjudication prompt format and examples in Section C.
>
> **Outcome (after rebuttal).**
>
> - In a detailed post-rebuttal comment, **qif8 writes that the updated version "significantly improves clarity"**, explicitly noting that definitions (including $N$, $d$), plan enumeration/instantiation, and calibration are now much clearer.
> - They keep a rating of 4 mainly due to their view on theoretical novelty and code availability at review time, but explicitly state that they **"would not oppose another reviewer championing the paper."**

---

> > ### Author Response · Authors · 2025-12-02
> > **Summary for the new Area Chair (Part 3)**
> >
> > ### Reviewer XCRa  (rating **4**, confidence as in review)
> >
> > **Initial concerns.**
> >
> > - Questioned the **novelty of the method** relative to existing VSA/HDC work, wondering whether PathHD goes beyond "repackaging" known binding and similarity operations.
> > - Raised concerns about **system-level limitations**, including dependence on the plan stage, Top-K coverage, and reliance on the LLM adjudicator.
> > - Asked for more discussion of the **theoretical underpinnings and scalability** of the HDC retriever.
> >
> > **Rebuttal and changes.**
> >
> > - **System-level positioning (Sec. 1, Sec. 2, Sec. 5).**
> >   We now clearly state that PathHD’s main novelty is **not a new VSA primitive**, but the **KG–LLM system design**: replacing neural path scorers with an HDC retriever and showing that this yields comparable accuracy with much lower latency and memory, plus interpretable path rationales. We contrast this with StructGPT, RoG, Think-on-Graph, GoG, KG-Agent, and FiDeLiS, which all rely on multi-call LLM agents and/or neural scoring modules.
> >
> > - **Top-K pruning and coverage (Sec. 3.4, Section I.7).**
> >   We added experiments (Table 5, Sec. 3.4) analyzing the impact of Top-K on accuracy and latency and showing that small K (2–3, default K = 3) retains or slightly improves accuracy while reducing latency. Section I.7 discusses coverage of gold paths under our chosen beam width and K, and characterizes failure cases for extremely compositional queries.
> >
> > - **Theoretical guarantees and complexity (Sec. 2.7 & Appendix E).**
> >   We added a capacity bound showing that the probability of a false match decays exponentially with $d$, and a complexity analysis with retrieval cost $O(N d)$, clarifying the scalability of HDC retrieval in our setting.
> >
> > **Outcome (after rebuttal).**
> >
> > - XCRa did not post a further comment after the rebuttal, but their written review indicates that they **“would not mind if the paper is accepted,”** and the concerns above are directly addressed by the new positioning, Top-K / coverage analysis, and theoretical discussion.
> >
> > ---
> >
> > ### Reviewer Rc8u  (rating **4**, confidence as in review)
> >
> > **Initial concerns.**
> >
> > - Shared some of XCRa’s questions about **novelty and limitations**, and additionally focused on:
> >   - **evaluation practice and baselines** (reliance on reported numbers vs. own runs, SOTA wording), and
> >   - **reproducibility**, including availability of code and clarity of algorithmic details.
> >
> > **Rebuttal and changes.**
> >
> > - **Evaluation protocol and baselines (Sec. 3.2–3.3).**
> >   We clarify that we **reuse officially reported baseline numbers** for methods that also follow the Freebase+official-script protocol—this is **standard practice in KG–LLM and LLM/NLP work** to avoid noise from partial re-implementations. We also add our own runs for strong LLM baselines (e.g., GPT-4 / LLaMA-style) under exactly the same protocol and soften over-strong "SOTA" wording, emphasizing that our main claim is about **comparable accuracy with significantly better latency and memory**.
> >
> > - **Reproducibility enhancements (Section A, Sec. 2.4–2.6, Sections F–I).**
> >   We expanded the notation table (Section A), gave more detailed algorithmic descriptions and pseudo-code (Sec. 2.4–2.6, Algorithm 1), and centralized hyperparameter settings (Sections F–I). We reiterate that **full code and configuration files will be released upon acceptance**, to enable exact reproduction.
> >
> > - **Additional analyses (Sec. 3.4–3.5, Section I).**
> >   Dimension studies, Top-K analysis, case studies, and prompt-sensitivity experiments were added to make the system’s behavior more transparent and address concerns about robustness and limitations.
> >
> > **Outcome (after rebuttal).**
> >
> > - Rc8u did not add a post-rebuttal comment, but, like the other 4-score reviewers, states that they **"would not mind if the paper is accepted."** Their concerns on evaluation and reproducibility are addressed by the clarified protocol, additional baselines, and strengthened documentation.
> >
> > ---
> >
> > ### 4. Final remark
> >
> > Taken together, the post-rebuttal situation is:
> >
> > - one clearly positive reviewer (6) emphasizing the value of the system-level contribution,
> > - three reviewers at 4 who **would not mind acceptance**, with one of them explicitly stating they **would not oppose another reviewer championing the paper**, and
> > - **substantial revisions** that improve clarity, add theoretical and empirical analysis, and directly address the main concerns raised in the reviews.
> >
> > We hope this reviewer-specific summary helps contextualize the reviews and revisions, and we appreciate your consideration of PathHD in light of its strengthened clarity, experiments, and system-level contribution.
> >
> > Thank you very much for your time and for considering our work!
> >
> > Sincerely,
> >
> > Authors

---

### Note · Program_Chairs · 2026-01-17
**Submission Desk Rejected by Program Chairs**

The following references in this submission do not refer to real documents and/or have major errors in bibliographic information:

 Mohsen Imani, Saransh Gupta, Yeseong Kim, and Tajana Rosing. Adaptable hyperdimensional computing for efficient learning and inference. IEEE Transactions on Computers, 68(8):1175-1188, 2019a.